# The Role of Silver Nanoparticles in the Diagnosis and Treatment of Cancer: Are There Any Perspectives for the Future?

**DOI:** 10.3390/life13020466

**Published:** 2023-02-07

**Authors:** Peter Takáč, Radka Michalková, Martina Čižmáriková, Zdenka Bedlovičová, Ľudmila Balážová, Gabriela Takáčová

**Affiliations:** 1Department of Pharmacology and Toxicology, University of Veterinary Medicine and Pharmacy, Komenského 73, 041 81 Košice, Slovakia; 2Department of Pharmacology, Faculty of Medicine, Pavol Jozef Šafárik University, 040 01 Kosice, Slovakia; 3Department of Chemistry, Biochemistry and Biophysics, University of Veterinary Medicine and Pharmacy, Komenského 73, 041 81 Košice, Slovakia; 4Department of Pharmaceutical Technology, Pharmacognosy and Botany, University of Veterinary Medicine and Pharmacy in Košice, 041 81 Košice, Slovakia; 5Department of Dermatovenerology, Faculty of Medicine, Pavol Jozef Šafárik University, 040 01 Košice, Slovakia

**Keywords:** silver, nanomedicine, antitumor mechanisms, cancer therapy, cancer diagnosis

## Abstract

Cancer is a fatal disease with a complex pathophysiology. Lack of specificity and cytotoxicity, as well as the multidrug resistance of traditional cancer chemotherapy, are the most common limitations that often cause treatment failure. Thus, in recent years, significant efforts have concentrated on the development of a modernistic field called nano-oncology, which provides the possibility of using nanoparticles (NPs) with the aim to detect, target, and treat cancer diseases. In comparison with conventional anticancer strategies, NPs provide a targeted approach, preventing undesirable side effects. What is more, nanoparticle-based drug delivery systems have shown good pharmacokinetics and precise targeting, as well as reduced multidrug resistance. It has been documented that, in cancer cells, NPs promote reactive oxygen species (ROS) production, induce cell cycle arrest and apoptosis, activate ER (endoplasmic reticulum) stress, modulate various signaling pathways, etc. Furthermore, their ability to inhibit tumor growth in vivo has also been documented. In this paper, we have reviewed the role of silver NPs (AgNPs) in cancer nanomedicine, discussing numerous mechanisms by which they render anticancer properties under both in vitro and in vivo conditions, as well as their potential in the diagnosis of cancer.

## 1. Introduction

Cancer is a complex, multifactorial disease characterized by the uncontrolled growth and spread of abnormal cells due to a combination of genetic, external, internal, and environmental factors [1], and it is treated with a variety of therapies, including chemotherapy, hormone therapy, surgery, radiation, immunotherapy, and targeted therapy. Despite considerable research advancements, the battle against cancer remains one of the greatest problems facing the scientific community. Current cancer therapies have significant drawbacks, including undesirable side effects, poor selectivity, and drug resistance. Consequently, the challenge is to discover efficient, cost-effective, and sensitive lead compounds with cell-targeted specificity and to enhance their sensitivity.

Nanomedicine for cancer is emerging as a new research topic with the goal of developing nanoinstruments for oncological applications. The ultimate objective of cancer nanomedicine is to enable early tumor detection, accurate diagnoses, and individualized treatment. Utilizing nanoparticles that can function at the molecular level is the primary benefit of cancer nanomedicine on the path to more customized medicine. Nanoparticles are defined by the American Society for Testing and Materials (ASTM) E2456-06 standard as particles with a size between 1 and 100 nm. However, not all writers have precisely adhered to these criteria. Thus, according to some studies, particles bigger than 100 nm are nanoparticles (NPs) [2,3]. According to their composition, nanoparticles may be categorized as polymeric, metallic, magnetic, carbon nanotubes, liposomes, dendrimers, or quantum dots (QD) [4].

In recent years, several nanoparticles, including carbon nanotubes [5], paramagnetic nanoparticles [6], liposomes [7], gold nanoparticles [8], and many more [9], have been explored for cancer diagnostics and therapy. For instance, nanoparticles may be used to enhance the pharmacokinetic and pharmacodynamic characteristics of current anticancer drugs and cancer treatment combinations [10,11]. Several metallic nanoparticles have shown significant antitumoral action against human cancer cells [12] in this setting.

Due to their intriguing physical–chemical features, silver nanoparticles (AgNPs) are gaining increasing attention in cancer research, showing inherent antiproliferative action [13]. AgNPs were studied to build a cancer diagnostic and therapy tool of the next generation [14]. It is feasible to synthesize stable AgNPs in a straightforward and cost-effective manner using a “green” synthesis technique. Silver is a noble metal having desirable biological characteristics, such as antibacterial and antifungal properties. Recent research has shown that several silver compounds have multiple impacts on cancer cells [15]. Due to the excellent physiological system of detoxifying in the human body, silver demonstrates low toxicity but poor absorption. Therefore, AgNPs are an effective means of avoiding this issue. Indeed, cells may internalize AgNPs by endocytosis and other uptake processes, releasing silver’s reactive species, Ag+ ions, at the target locations [16].

Multiple pathways provide broad-spectrum anticancer action to AgNPs [17,18,19]. Numerous in vitro and in vivo research studies have shown that AgNPs may inhibit the growth and viability of cancer cells. By damaging the ultrastructure of cancer cells and causing the formation of ROS and DNA damage, AgNPs may induce apoptosis and necrosis [20,21,22]. AgNPs may induce apoptosis by up- or down-regulating the expression of important genes, such as p53 [23,24], and by altering crucial signaling pathways, such as the hypoxia-inducible factor (HIF) pathway [25]. AgNP-treated cancer cells may also exhibit cell cycle arrest [26,27]. Upon exposure to AgNPs, several cancer cells experience sub-G1 arrest and apoptosis. By inhibiting tumor cell migration and angiogenesis, AgNPs may also diminish distant metastasis [28,29]. To produce a safe and efficient anticancer drug, other mechanisms for the anticancer actions of AgNPs must be investigated.

The aim of this review is to summarize the in vitro and in vivo anticancer mechanisms of AgNPs, as well as their potential in cancer diagnosis.

## 2. The Potential of AgNPs in Nano-Oncology

AgNPs have shown impressive capabilities and promising outcomes in cancer diagnostics and therapies. The drugs currently used for anticancer treatments are toxic to the body, causing side effects and unintended or untargeted effects on normal body physiology, as well as the development of drug resistance, rapid drug metabolism, and clearance from the patient’s body, which decreases effective treatment time. The biological production of metal nanoparticles yields safe and potent anticancer medicines. NPs synthesized with different metals may exhibit distinct characteristics and, thus, are anticipated to exhibit distinct modes of toxicity against diverse cancer cells. To combat cancer cells, AgNPs may also be conjugated or combined with medicines or coated with a polymer.

In the next part, we point out the potential of AgNPs in nano-oncology, both from the point of view of diagnostic and therapeutic use. We focus on the mechanisms of antitumor activity under both in vitro and in vivo conditions.

### 2.1. The Role of Silver Nanoparticles in the Diagnosis of Cancer

Personalized cancer medicine provides enormous promise for theranostics combined with diagnosis and treatment on a single nanoplatform [30,31,32]. In contrast, traditional theranostic nanoplatforms often combine therapeutic drugs with imaging probes through encapsulation or other means. In this manner, both the diagnostic and therapeutic processes are presented as an “always-on” mode [33], in which the theranostic agents can function without exogenous or endogenous stimuli, resulting in a limited signal-to-noise ratio (SNR) in the disease sites and an inevitable side effect on normal cells. Recent research has been focused on activatable theranostic platforms, in which the biomarkers that may trigger the diagnosis and treatment are essential for distinguishing normal cells from cancer cells.

In addition, it is hoped that novel tactics integrating the use of NPs would facilitate the early identification and diagnosis of cancer, a crucial factor in the final success of cancer therapy. Early diagnosis of tumors in patients facilitates cancer management and limits its spread and is crucial for determining future prognosis. Thus, techniques that allow for the identification and imaging of tumors are crucial. Due to their well-known high absorbance and light scattering in the plasmon resonance wavelength areas [34], both AuNPs and AgNPs have been exploited in cell imaging applications.

Due to the higher levels of toxicity associated with AgNPs, AuNPs tend to be employed to a larger degree for this purpose. In a recent work, SERS-encoded magnetic AgNPs as multifunctional tags for cancer cell targeting and separation were synthesized and evaluated [35]. The magnetic core and silica coating of AgNPs conferred chemical and physical stability. Strong SERS signals generated by silica-coated-AgNPs were discovered to be responsible for the effective targeting of SKBR3 breast cancer cells and SP2/O leukemia cells. In addition, an external magnetic field permitted the separation of targeted cancer cells from nontargeted cells.

The use of AgNPs to treat triple-negative breast cancer is an example of their theranostic use (TNBC). Due to the aggressive nature of the tumor, delayed detection, and the lack of identifiable symptoms in the early stages, this illness is linked with the poorest clinical results. Consequently, the development of novel blood biomarkers unique to TNBC for diagnostic and therapeutic reasons remains an important clinical necessity. Blood-dispersed silver nanoparticles are encased in a unique protein covering, the so-called “Protein crown” (PC). Changes in protein patterns are difficult to detect with traditional blood tests; nevertheless, PC functions as a “nanoconcentrator” of serum proteins with affinity for the surface of AgNP [36]. Thus, characterization of the PC could enable the detection of otherwise undetectable changes in protein concentration at an early stage of the disease or after chemotherapy or breast surgery using liquid chromatography methods, tandem mass spectrometry (LC-MS/MS), and confirmation via a sequence window acquisition of all theoretical mass spectra (SWATH) with an AgNP size of approximately 10 nm. Interacting with human blood serum, these AgNPs may aid in the diagnosis of other civilization-defined disorders [37].

In the field of biomedical applications, research into the design and delivery of NPs to a specific organ has grown. To determine if nanoparticles have been transported to sick tissue and to determine the intended function of NPs, a novel combination of photoacoustic imaging and custom-designed nanosystems [38,39] has been developed. First, the item will absorb light, then the absorbed light energy will be converted into heat, and finally, due to thermoelastic expansion, sonic waves will be released [40]. Oraevsky was the first to propose photoacoustic imaging for biological purposes [41,42]. AgNPs are used as contrast agents for imaging therapy because of their high optical absorbance and scattering capabilities. Ultrasound imaging and photoacoustics could detect NPs injected 1 cm deep into ex vivo pancreatic tissue using 800 nm wavelength light and radiation. The first way is that AgNPs are tailored to target the tumor site, specifically at the leaky blood vessels of the tumor, as well as the decreased rate of clearance owing to the absence of functioning lymphatic vessels, and the AgNPs will be retained. At the location of the tumor, AgNPs are conjugated with antibodies and bind to the antigens present. By localizing the AgNPs to the location of the tumor, it offers photoacoustic contrast with normal tissues, making it effective for the in vivo assessment of tumors.

Exosomes positive for the prostate-specific membrane antigen (PSMA) have the potential to function as very sensitive biomarkers for the diagnosis of prostate cancer. Based on a peptide-templated AgNPs nanoprobe, a sensitive electrochemical biosensor for the ultrasensitive detection of PSMA-positive exosomes has been developed. PSMA-specific binding peptides mounted on a gold electrode were responsible for trapping exosomes produced from prostate cancer in this study. A well-designed peptide (CCY-LWYIKC) serves two purposes: as a signal probe and an exosomes identification recognizer. Specifically, LWYIKC bind cholesterol at exosome membranes, while CCY serve as peptide templates to host a high number of silver nanoparticles, resulting in a robust electrochemical signal. This approach demonstrated exceptional efficacy when evaluated using clinical samples, indicating its potential for clinical applications [43].

In the work of Cheng et al. (2023), a high-sensitivity thermally annealed silver nanoparticle/porous silicon bragg mirror (AgNPs/PSB) composite substrate was used to boost the Raman spectroscopy (RS) signal of serum using the surface-enhanced Raman spectroscopy (SERS) approach. SERS reflects greater and stronger spectral peak information than RS, which is advantageous for discovering novel breast cancer biomarkers. The experimental findings indicate that the breast cancer detection model based on the enhanced SERS substrate and the machine learning algorithm may be utilized to differentiate breast cancer patients from healthy controls. The aforementioned experimental findings indicate that the SERS technology based on an AgNPs/PSB composite substrate, when paired with machine learning techniques, offers tremendous promise for the speedy and accurate diagnosis of breast cancer patients [44].

### 2.2. Mechanisms of Antitumor Efects of AgNPs In Vitro

The goal of antitumor therapy is to inhibit the proliferation of tumor cells and induce their death without having a significant negative impact on the surrounding healthy tissue. Conventional therapy involves the use of molecules that are able to intervene in the regulation of the cell cycle, inhibit proliferation, and have a cytotoxic effect, which is the cause of the occurrence of unacceptable side effects [45]. Moreover, many types of malignant tumors are resistant to this treatment [46]. For this reason, due to their unique physicochemical properties, AgNPs have become a new hope in the treatment of cancer. By targeting drug delivery directly to cancer cells, their better bioavailability can improve the effectiveness of antitumor therapy and minimize side effects [47]. The mechanisms of action of silver nanoparticles and their conjugates with antitumor agents include the production of ROS and oxidative stress, DNA damage, cell cycle arrest, and the induction of tumor cell death by apoptosis, as well as by nonapoptotic types of cell death [48,49,50].

#### 2.2.1. Effect of AgNPs on Cell Cycle Regulation

Cell cycle progression is an essential mechanism that ensures DNA replication, cell growth and division, and organism renewal. However, the overexpression of growth factors or mutations in genes/lack of controlling tumor suppressor proteins leads to the inability of cells to exit the cell cycle and subsequent uncontrolled cell division. For this reason, modulation and targeting of cell cycle control mechanisms represent a suitable therapeutic target for antitumor therapy [51]. The cell cycle is strictly regulated in healthy cells by several proteins such as cyclins, cyclin-dependent kinases (CdK), and CdK-activating kinases (CAKs). Topoisomerases, tubulins, various enzymes, and proteins from the family of cyclin-dependent kinase inhibitors also participate in the progression of the cell cycle. DNA damage can lead to irreversible cell cycle arrest at various phases of the cycle, which is directly linked to the activation of a complex machinery resulting in cell death [52,53].

Although numerous studies indicate that AgNPs are capable of arresting the cell cycle in different phases, the most common occurrence is the accumulation of tumor cells exposed to AgNPs in the G2/M phase [54,55]. Their ability to cause DNA double-strand breaks and increase the population of AgNPs-treated cells with subG0/G1 DNA content, which are considered apoptotic, is also known [56]. Several studies also point to the influence of AgNPs on the expression of regulatory proteins involved in the modulation of the cell cycle (Table 1). Such important transcription factors with a proapoptotic function include p53, the best-known tumor suppressor, which is involved in intracellular processes such as the response to DNA damage and its repair, regulation of cell metabolism and autophagy, aging, and cell death [57].

Noorbazargan et al. (2021) [29] studied the antitumor effects of silver nanoparticles (used at a concentration of 9.87 µg/mL) prepared using *Juniperus chinensis* extract on human lung cancer (A549) tumor cells. Cells were exposed, and their effect was compared with that of cisplatin (CisPt) used at a concentration of 24.67 µg/mL, and it was shown that NPs more effectively induced an increase in p53 expression than CisPt. This effect is thought to be due to a response to increased levels of nuclear fragmentation, disintegration, and chromatin condensation, which may have been induced by increased ROS production. Thus, activated p53 affects the transcription and function of genes necessary for the progression of the cell cycle, which was proven by a significant decrease in the level of cyclin D1 with the simultaneous accumulation of cells in the G0/G1 phase, which resulted in the induction of apoptosis. Apoptosis was confirmed by the upregulation of initiator caspase -9 and executioner caspase -3. Moreover, NPs significantly reduced the expression of matrix metalloproteinases MMP-2 and MMP-9, which are closely related to angiogenesis, invasiveness, and metastasis [58].

Similar mechanisms were also described on human breast adenocarcinoma (MCF-7) cells after exposure to AgNPs prepared using *Bergenia ligulata* aqueous extract (at a concentration of 5 and 10 μg/mL). Increased activation, i.e., upregulation of p53 phosphorylation led to the arrest of the cell cycle in the G2/M phase and, together with oxidative stress mediated by ROS generation, probably participated in the induction of apoptosis. Apoptosis was demonstrated by changes in the levels of proapoptotic and antiapoptotic factors such as Bax and Bcl-2, reduction in mitochondrial membrane potential (MMP), and activation of caspase -3 [59]. The effect of polyphenols from the extract of the lichen *Parmelia sulcata*, with the help of which AgNPs were prepared at a concentration of 30 µg/mL, was studied on the same cells. It was found that in addition to the downregulation of cyclin D1, which is essential for the progression of the cell cycle, there was also a decrease in the intracellular levels of PCNA (proliferating cell nuclear antigen), which is an important cofactor of DNA polymerase δ involved in DNA replication and repair. In addition, the expression of inflammatory genes such as TNF-α and IL-6 was suppressed [60].

A significant reduction in the expression of cyclins B, E, and D was also demonstrated after the action of AgNPs synthesized using the *Annona muricata* plant on A549 cells (human lung carcinoma). In addition to the upregulation of p53, increased production of ROS and proapoptotic factors such as Bax, caspases, and p21 protein, another important tumor suppressor, was also upregulated [61]. Decreased expression of cyclin D1 and increased expression of both p21 and p27 were also observed in human prostate tumor cells (DU145) treated with AgNPs synthesized using *Carica papaya* leaf extract. Protein p21/Cip is a universal inhibitor of all cyclin/CdK complexes. It is an important tumor suppressor protein whose activity is stimulated by p53 in response to DNA damage. Similarly, p27/Kip, which belongs to the family of Cip/Kip proteins, prevents the activation of the cyclin D-CDK4 complex and cell cycle progression. These effects led to cell cycle arrest in the G2/M phase and the induction of apoptosis [62].

Hembram et al., in 2020 [63], evaluated the antitumor activity of hybrid Quinacrine-Based silver and gold nanoparticles on various tumor and nontumor cell lines (A549, A431, HEK-293, HeLa, HCT116, MCF-7, H-357, SSC-25, SCC-9, and MCF-10A), and the minimum inhibitory concentration values were in the range of 0.5–27 µg/mL. On SCC-9 (squamous cell carcinoma) cells, they studied the effect of NPs on changes in the levels of many proteins involved in DNA repair, replication, and cell cycle regulation. They noted a significant reduction in the expression of cyclins E1, B1, and A2. Cdc-2 (Cdk1)/Cdk2 and cyclin A2 complexes participate in the transition of cells from the S phase to the G2 phase of the cell cycle. The activity of these complexes is negatively affected by Cip/Kip proteins such as p21 and p27 and checkpoint kinases (Chk1) [64], whose levels were significantly reduced after NPs administration (Chk2 was paradoxically upregulated, probably due to the activation of a compensatory mechanism). Cdks are the downstream target of Cdc25-A phosphatase, which activates them and maintains cell cycle progression, which was also reduced after NP application. All the mechanisms mentioned, together with affecting the levels of other proteins involved in DNA repair, contributed to the inhibition of tumor cell growth and led to a cell cycle block in the S phase.

The involvement of intracellular signaling pathways with different functions also proves that the cell cycle block is influenced by complex mechanisms. PKC family proteins are known to be involved in cell growth and survival, as well as apoptosis and cell cycle regulation [65]. These assumptions were confirmed by a study on A549 lung cancer cells that were exposed to AgNPs. NPs showed a time- (12, 24, 48, and 72 h) and dose- (10–200 µg/mL) dependent antiproliferative effect with cell cycle arrest in the G2/M phase. This effect was mediated by upregulation of p21 and p53 with concomitant downregulation of PKCξ. Overexpression of this PKC isotype resulted in decreased sensitivity to the effect of NPs and increased tumor cell proliferation, suggesting that suppression of PKCξ expression is closely related to cell cycle arrest in the G2/M phase after AgNPs treatment [66].

These findings indicate that intervention in the regulation of the cell cycle through AgNPs (Figure 1) has a significant impact on their possible antiproliferative and antitumor effects, and therefore it is desirable to study and expand knowledge about their ability to interact with other important proteins involved in cell proliferation and survival. Below, we discuss the effect of NPs on p53 regulation and the induction of tumor cell death.

#### 2.2.2. AgNPs as Cell Death Inducers

Cell death is a complex process that regulates homeostasis and biochemical changes in healthy cells and throughout the organism. Part of conventional antitumor therapy is to affect the intracellular mechanisms of tumor cells as selectively as possible, resulting in their death. In connection with the induction of cell death, apoptosis and autophagy are currently intensively discussed; the simultaneous modulation of which can contribute to the increased effectiveness of antitumor treatment [67].

##### Apoptosis

Despite the constant discovery of still unexplained mechanisms of action of antitumor agents and the appearance of tumors with the presence of resistance to apoptotic stimuli mediated by various mutations, apoptosis remains a type of cell death, the induction of which is considered an essential mechanism of the antiproliferative action of these drugs in in vitro and in vivo conditions [68,69]. Targeting research on apoptotic cell death and its selective activation in tumor cells is a suitable strategy for the search for new drugs that could become more universal in the therapy of several types of cancer [70]. Apoptosis is an extremely complex, strictly organized process regulated by the balance between proapoptotic and antiapoptotic factors. Apoptosis can be triggered by an intrinsic, mitochondrial-dependent pathway, which depends primarily on irreparable DNA damage, activation of the p53 signaling pathway, eventual cell cycle arrest, and mitochondrial damage, which are essential for cell survival. Another pathway that leads to the induction of apoptosis is extrinsic, death-receptor-mediated cell death, which, like the mitochondrial pathway of apoptosis activation, leads to the triggering of proteolytic mechanisms through activation cleavage by caspases [71] (Figure 2).

The wide range of mechanisms of action of silver nanoparticles also includes the activation of several pathways of apoptosis (Figure 2). Studies point to the ability of AgNPs to induce the extrinsic pathway of apoptosis, which consists in the interaction between the death receptor (Fas, TNFR1, TNFR2, and the TRAIL receptors DR4 and DR5) and corresponding ligands (FAS, TRAIL or TNF). Subsequently, initiator caspases -8 and -10 are activated, and the proapoptotic stimulus is propagated [72]. AgNPs prepared using *Zizyphus mauritiana* fruit extract demonstrated exceptional antimicrobial, antifungal, and antiproliferative properties. Hormone-dependent breast adenocarcinoma (MCF-7) cells were exposed to NPs with an average size of 16 nm, and the IC50 was determined to be 28 μg/mL. Microscopic changes, increased ROS levels, and inhibition of colony formation indicated the involvement of one of the apoptosis pathways. The results show that there was a significant upregulation of FAS and FADD in cells treated with NPs, which are considered to be key in the induction of the extrinsic apoptotic pathway. FAS, belonging to a family of membrane proteins that are used by T lymphocytes and NK cells to destroy target tumor cells, and FADD (Fas Associated protein with Death Domain) are key proteins ensuring the recruitment of initiator caspases −8 and −10 [73,74]. The increase in caspase −8 expression initiated the activation of executioner caspase -3 and the reduction in uncleaved PARP, which are significant features of apoptosis [75]. AgNPs prepared from *Balanites aegyptiaca* aqueous extract upregulated the proapoptotic protein BIM from the Bcl-2 protein family, capable of interacting with antiapoptotic Bcl-2 proteins and the anti-inflammatory IκBα in Caco-2 colorectal cancer cells [76]. For the induction of the extrinsic pathway of apoptosis, AgNPs can also be used as particles that potentiate the effect of other substances. For example, conjugates of AgNPs and TRAIL (tumor necrosis factor-related apoptosis-inducing ligand), belonging to the TNF family, significantly sensitized TRAIL-resistant human glioblastoma cells and increased the activity of caspases −3/−7, −8 and −9 [77].

A much more studied mechanism in connection with AgNPs is the induction of the mitochondrial apoptotic pathway [78]. Numerous studies have shown that silver nanoparticles have a significant effect on changes in the levels of proteins from the Bcl-2 family, the function of mitochondria, and the activation of caspases. The primary trigger of this pathway is DNA damage, its fragmentation and chromatin condensation are probably associated with ROS production. This effect was demonstrated by AgNPs conjugated with *Rubus fairholmianus* extract on MCF-7 cells, synthesized from *Juniperus chinensis* extract on A549 cells, root extracts of *Beta vulgaris* L. on HuH-7 cells [29,79,80] biosynthesized *Catharanthus roseus* with an effect on HepG2 cells [81], or prepared using an ethanolic extract of *Calotropis gigantea* latex studied on EAC cells (Ehrlich-Lettre ascites carcinoma) [82]. If, after DNA damage, there is no repair and renewal of the cell cycle, the p53 protein as a transcription factor activates the expression of proapoptotic proteins. These include members of the Bcl-2 family with a proapoptotic effect such as Bax and BH3-only proteins, for example, Bim, Bid, PUMA, Noxa, and Bad [83]. In the study we mentioned above [63], AgNPs downregulated proteins such as Fen-1, XRCC-1, Pol-β, Pol-ε, DNA PKcs, NFКβ, Topo II α, MRE11, RFC-1, PCNA, helicase, and RPA involved in DNA repair and replication. This significant DNA damage led to the upregulation of histone γ-H2AX, a marker typical of DNA damage, as a result of which activated p53 promoted the apoptotic process. Another important step after p53 activation is the increase in the expression of Bax and related proteins. Bax is able to interact with the mitochondrial outer membrane and form pores that result in irreversible mitochondrial damage [84]. After depolarization of the outer mitochondrial membrane, other proapoptotic factors such as cytochrome c and Smac/DIABLO are released, with subsequent activation of the apoptosome [85]. Next, we describe a complex of mechanisms by which silver nanoparticles can contribute to the induction of apoptotic cell death. The results of the study from the last 5 years are summarized in Table 1.

Increasing the expression of proapoptotic Bax and PUMA (the p53-upregulated modulator of apoptosis) [86] and decreasing the expression of antiapoptotic Bcl-2 and Bcl-xL [82,87,88] disrupted the ratio of these members of the Bcl-2 protein family, leading to the induction of the mitochondria-mediated apoptosis pathway. The alteration in the activity of these proteins leads to mitochondrial dysfunction resulting in impaired ATP synthesis [89]. In addition to mitochondrial damage through oxidative stress and influencing Bcl-2 proteins, some believe that AgNPs are able to directly interact with mitochondria and that their effect may be particle-specific [90] or affect mitochondrial enzymes necessary for their proper function. Low doses of fructose-coated silver NPs inhibited PDK (pyruvate dehydrogenase kinase), which is a negative regulator of PDH (pyruvate dehydrogenase), leading to decreased glycolysis and increased glucose oxidation in osteosarcoma cells but not in healthy cells. This explains the selectivity of these AgNPs and the preferential production of ROS and induction of apoptosis in cancer cells [91].

The result of such mitochondrial damage by AgNPs is an increase in the number of cells with reduced MMP (mitochondrial membrane potential) [79,92], which could result in a decrease in the fraction of mitochondrial cytochrome c, with a simultaneous increase in cytochrome c levels in the cytosol [93] and the release of Smac/DIABLO [94], known as the second mitochondria-derived activator of caspase/direct inhibitor of apoptosis-binding protein with low pI. AgNPs, in which plant extracts and solutions were not used, are also involved in the upregulation of this important marker of the mitochondrial apoptosis pathway [95]. Activation of initiator and executioner caspases, which were activated after treatment of several types of tumor cells, occurs after mitochondrial damage. These mainly included caspases −3, −7, −8, and −9 [79,95,96,97,98,99,100].

The induction of apoptosis is also closely related to interference with the formation and function of the microtubule system, which is essential for cell life and cell division. Polymerization of tubulin dimers is an energy-demanding, ATP-dependent process, the disruption of which can cause cell damage and/or death. During the execution phase of apoptosis, the formation of the AMN (apoptotic microtubule network) can even occur [101]. AgNPs can also intervene in this process with their ability to induce the disruption of tubulin fibers, shrinking of cell nuclei, and reduction in the cytoplasmic volume of human osteosarcoma cells [102], even inducing alterations in the integrity and organization of the cytoskeleton and impairment of cell membrane structures, which lead to the death of cells [103].

Activation of caspases leads to proteolytic cleavage of intracellular molecules. The substrate for effector caspases is also PARP (Poly (ADP-ribose) polymerase), an important DNA repair enzyme ensuring the stability of the genome. Its cleavage disrupts its functionality, and therefore the presence of a cleaved form of PARP is considered a marker of apoptotic cell death [104]. Several AgNPs are able to increase PARP cleavage, for example, in colorectal cancer (HCT116), lung cancer (A549) or human squamous cell carcinoma (SCC-9) cells [63,86,94].

Activation of the extrinsic and intrinsic pathways of apoptosis is the most studied mechanism associated with the antiproliferative and antitumor effects of potential drugs, but there are other intracellular processes contributing to the induction of apoptosis. Endoplasmic reticulum (ER) stress caused by nutrient deficiency, the action of drugs, oxidative stress, and other factors such as the expression of mutated and misfolded proteins leads to the activation of the unfolded protein response (UPR), which is an important protective mechanism leading to cellular homeostasis. However, prolonged ER stress leads to apoptotic cell death [105]. The physiological UPR reaction is ensured by three key transmembrane proteins: IRE1 (inositol-requiring enzyme 1), PERK (protein kinase RNA-like ER kinase), and ATF6 (activating transcription factor 6), considered sensors of ER stress, which are kept inactive in nonstress conditions [106]. Several studies indicate that the induction of ER stress and changes in the levels of these proteins may be an important mechanism of action of AgNPs as antitumor drugs [107] (Figure 2).

The study on drug-resistant MCF-7/ADR cells treated with theranostic nanocomposite prepared using AgNPs and peptide-functionalized doxorubicin, whose concentration corresponded to 50 μg/mL of doxorubicin, brought remarkable results. It was found that these NPs were able to activate the PERK-mediated signaling pathway. Activated PERK phosphorylates eIF2α and activates downstream target proteins such as ATF4 and CHOP/GADD153 (growth arrest and DNA damage 153), which as a transcription factor regulates the expression of pro- and antiapoptotic proteins. Under stress-free conditions, PERK is bound by GRP78/BiP on the ER membrane, but ER stress facilitates the dissociation, phosphorylation, and oligomerization of PERK [108]. Activation of this signaling pathway resulted in an increase in cytoplasmic calcium levels and activation of caspase -12 [109]. Similar effects were also reported on the human breast cancer cells MDA-MB-231 and MCF-7 [110,111]. In addition, AgNPs also activate the IRE1 (endoribonuclease) branch of the UPR, which results in the cytosolic splicing of XBP1, a transcription factor that can subsequently translocate to the nucleus. ER stress results in the activation of caspases as well as noncaspase proteases, for example, calpain [21]. ER stress and UPR may also be associated with changes in P-gp levels and functionality [109,112].

It follows from the above that the induction of apoptosis by the extrinsic, intrinsic or ER-mediated pathway is an extremely important mechanism of action of potential antitumor drugs. However, the mechanism of the antitumor effect of AgNPs also involves triggering nonapoptotic types of cell death.

##### Nonapoptotic Cell Death

Programmed cell death (PCD) includes apoptosis and nonapoptotic types of cell death such as autophagic cell death, pyroptosis, necroptosis, ferroptosis, and others. Apoptosis-independent cell death is a suitable target for new antitumor treatment, especially given the ever-increasing resistance of tumor cells to apoptosis and the currently used mechanisms of antitumor therapy. Due to their unique properties, nanoparticles represent a new perspective on the possibilities of bypassing apoptosis, both as carriers of bioactive substances but also as independently effective substances, either in monotherapy or in combination with classical antitumor treatments such as photothermal, photodynamic therapy, immunotherapy, and others [90,113].

##### Autophagy and Autophagic Cell Death

Autophagy, also known as macroautophagy, is a significantly important process contributing to the survival of cells under stressful conditions, especially in the case of nutritional deficit, oxidative stress, low intracellular ATP levels or hypoxia. Its task is the degradation of nonessential cellular macromolecules, protein aggregates, and even entire damaged intracellular organelles for the purposes of nutrient recycling, energy generation, and cell survival. Such degradation is also used in the elimination of oncogenic molecules, and thus autophagy has a tumor-suppressive effect, but if it is activated in cancer cells, it can contribute to the survival of cancer cells and tumor resistance to treatment. Autophagy is considered a double-edged sword in cancer and, therefore, understanding the mechanisms involved in the autophagic process, its connection to other processes such as apoptosis, endoplasmic reticulum stress, p53-associated signaling pathways, and many others is essential for the effective use of autophagy as an antitumor mechanism [114,115].

Some studies suggest the role of AgNPs as autophagy inducers with a protective effect (Figure 3). To clarify this hypothesis, it is possible to use the known autophagy inhibitor chloroquine or to test the effect of AgNPs in hypoxic conditions in which autophagy induction was assumed [25,115,116]. On the other hand, many studies confirm that there are many nodes of crosstalk between autophagy and apoptosis and that their simultaneous activation acts synergistically and potentiates the mutual antitumor effect [117].

The key sensor regulating cellular metabolism and homeostasis is AMPK (AMP-activated protein kinase). It is known that changes in the ratio of ATP and AMP increase its activity with subsequent phosphorylation of the mammalian autophagy-initiating kinase Ulk1, a homologue of yeast ATG1, which is necessary for the formation of the initiation complex mediating the next steps of ongoing autophagy. On the other hand, the most studied antagonist, i.e., negative regulator of ULK1, is mTOR (mammalian target of rapamycin), a central molecule crucial for the proliferation, growth, and survival of tumor cells [118].

Chen et al. (2020) [119] used PVP-modified AgNPs to study NPs-mediated autophagy in human prostate cancer (PC-3) cells at sublethal doses. Flow cytometric analysis showed that after AgNPs application, apoptosis did not occur, but all autophagy markers were significantly upregulated. This effect was associated with lysosome damage, a decrease in their number with a decrease in protease activity, and a significant downregulation of cathepsin D, which led to the blockade of autophagic flux. Hypoxia (significant upregulation of HIF-1) and energy deficit induced by the action of NPs led to the activation of AMPK and the suppression of mTOR expression and phosphorylation, with subsequent downstream activation of the AMPK substrate ACC1 (acetyl-CoA carboxylase1) and suppression of the expression of the mTOR target protein S6K (ribosomal protein S6 kinase beta -1), which regulates protein synthesis, growth, and the proliferation of cells and their survival. The LC3-I/II protein is considered to be the most important marker of autophagy, the elevation of which indicates ongoing autophagy. Basic indicators also include the p62/SQSTM1 (sequestosome 1) protein, an autophagic receptor essential for the recruitment of cytoplasmic cargo destined for degradation [114]. Considering that after the treatment of tumor cells there was a significant accumulation of p62, the authors believe that a blockade of the autophagic flux probably occurred. In this case, AgNPs activated autophagy as a protective mechanism and, therefore, its inhibition could increase the antiproliferative effect of silver nanoparticles.

In contrast, AgNPs biogenerated by *Klebsiella oxytoca* under anaerobic conditions induced autophagic cell death in SKBR-3 (human breast adenocarcinoma) cells [120]. Compared with the above-mentioned study, the autophagic flux was fully developed, which was confirmed primarily by a significant downregulation, i.e., consumption of p62 with a simultaneous increase in Beclin-1 levels. Beclin-1 is part of PI3K-III complex 1 and is considered a central molecule regulating autophagy and a crosstalk point between autophagy and apoptosis. Cleavage of Beclin-1 by caspases or calpains results in the predominance of apoptosis and suppression of autophagic flux [121]. Decreased expression of both total and phosphorylated Akt suggests inhibition of the PI3K/Akt/mTOR signaling pathway, which acts as a suppressor of autophagy initiation. At the same time, other important proteins such as ATG5 and ATG7, as well as the increased conversion of LC3-I to LC3-II, were also upregulated. The low participation of apoptosis in the death of tumor cells (only 11.57%) compared with untreated cells (5.5%) without the presence of DNA fragmentation and, at the same time, the significant presence of lysosomes, autophagolysosomes, and acidic vesicular organelles indicates that autophagy had a dominant position as a type of cell death induced by AgNPs in tumor cells [120]. Even polydopamine-coated Au–Ag nanoparticles induced mitochondrial damage and lysosomal membrane permeability, multiple cell death pathways, dose-dependent apoptosis, necrosis, and autophagy after irradiation [122]. A similar transition from apoptosis to autophagy was observed in HT-29 colorectal cancer cells. Upregulation of apoptosis-mediating factors such as p53, cytochrome c, Bax, and caspase -3, -8, and -9, as well as an increase in the levels of autophagic proteins (Beclin-1, LC3-II), were noted. Proteins such as ATG3 and ATG12 were downregulated. The authors believe that, although these proteins are necessary for the induction of apoptosis, endoplasmic reticulum stress and UPR mediated by AgNPs exposure and demonstrated by the upregulation of CHOP and XBP1 induced late noncanonical autophagy [123]. Autophagy can also be triggered by various other signaling pathways. Its induction is mediated by reducing the expression of its inhibitory regulator, such as Akt/mTOR signaling, but also by inhibiting the NF-κB signaling pathway [124].

All the above-mentioned results of studies from recent years indicate a huge complex of mechanisms by which AgNPs can induce the death of tumor cells. However, less known types of cell death have also been studied [113].

Zielinska et al. (2018) [125] studied the mechanisms of induction of cell death induced by AgNPs on pancreatic ductal adenocarcinoma (PANC-1) cells, and their results were surprising. The used uncoated NPs simultaneously induced a mixed type of programmed cell death including apoptosis, autophagy, necroptosis, and mitotic catastrophe. Using TEM (transmission electron microscopy), they monitored morphological changes in cells treated with AgNPs after administration at different doses and particle sizes. The apoptosis present was accompanied by typical nuclear and cytoplasmic condensation and blebbing, and Western blot analysis showed increased expression of p53, Bax, and downregulation of Bcl-2. Autophagy was presented by the upregulation of LC3-II and the presence of autophagovacuoles and autophagosomes with the presence of AgNPs inside. Upregulated proteins also included RIP-1, RIP-3, and MLKL, which are typical markers of ongoing necroptosis. Their activation leads to membrane pore formation and a massive influx of calcium, resulting in cell death [126]. Necroptotic cell death was accompanied by intense cytoplasmic vacuolization, dilation of the nuclear membrane, and disruption of the cell membrane. In addition, the presence of signs of mitotic catastrophe—increase in cell size, micronucleation, and swirling of mitochondrial cristae—was detected. The antiproliferative effect of these AgNPs was more selective toward tumor cells compared with a nontumor cell line. Silver NPs also activated a paraptosis-like mixed type of cell death, which is characterized by the involvement of MAPK signaling, ROS production, and extensive cytoplasmic vacuolation due to dilation of the ER, Golgi apparatus, and mitochondria. The influence of p62 and LC3-II proteins also indicates the involvement of autophagy, and the increase in MLKL phosphorylation is a marker of necroptosis. DNA fragmentation and caspase-3 activation were not significant; therefore, the authors assume that apoptosis was not the main cell death mechanism of tumor pancreatic cells [127].

The induction and type of programmed cell death also depend on the type of tumor cells, mutation, and tumor microenvironment. The same AgNPs induced pyroptosis of MDA-MB-231 and LNCaP cells, necroptosis of MDA-MB-231 cells, and apoptosis of HCT116 cells [128].

All these results prove that it is necessary to pay attention to other types of cell death as a possible mechanism of antitumor and antiproliferative effect of potential antitumor drugs.

Cell death is accompanied by changes in the regulation of various signaling pathways, and therefore we continue to describe the signaling pathways modulated by AgNPs (Figure 3).

#### 2.2.3. AgNPs and MAPK Pathways

The physiological function of members of the MAPK (mitogen-activated protein kinase) family is the ability to transmit extracellular signals and mediate intracellular responses. Their role is to amplify such signals and contribute to cell growth, proliferation, and survival. On the other hand, their extensive activation is often found in many types of tumors, as their overexpression supports tumor proliferation, invasiveness, metastasis, angiogenesis, and resistance to anticancer treatment. Several cascades belonging to this family have been identified, including ERK1/2, c-Jun N-terminal kinase (JNK), and p38 MAPK [129,130].

In this context, the impact of silver NPs on the expression of some members of these cascades was studied [131,132]. AgNPs prepared using sinigrin displayed a proapoptotic effect accompanied by the release of cytochrome c, activation of caspases (−3, −6, and −8), and upregulation of p21, p53, and proapoptotic proteins such as Bid, Bax, and Bak. The authors assume that apoptosis was triggered with the involvement of stress kinases p38 and JNK, which sensitively respond to stress conditions such as the presence of cytokines, osmotic stress, UV and gamma radiation, and others. The expression of both of these kinases and NF-κB was upregulated in HeLa cells treated with AgNPs, even in combination with camptothecin. On the other hand, there was a downregulation of Akt and members of the Raf-MER-Erk cascade. The suppression of the expression of kinases involved in growth-factor-associated growth and survival of cells likely contributed to the reduction In the survival of tumor cells and the activation of apoptosis [23]. Liu et al. (2021) [133] observed similar changes in the phosphorylation of p38, JNK, and Erk1/2 in U251 cells (human glioma). After the application of GSH (an antioxidant), they recorded a decrease in the activation of these kinases, and therefore, the authors point to the involvement of ROS in the activation of MAPK signaling pathways. Increased phosphorylation of Erk1/2 is generally considered to increase support for cell survival and proliferation; on the other hand, Erk-dependent mechanisms of apoptosis induction have been described [134].

These findings are supported by the results of other authors who observed a decrease in Akt and Erk1/2 levels in human bladder cancer cells (T24) after treatment with Au–Ag nanoparticles [135], activation of the p38 stress kinase in A549 cells (human lung cancer) induced by the generation of ROS with subsequent activation of caspase-3 [136], and an increase in the phosphorylation of JNK with upregulation of its direct downstream target c-Jun in various tumor cells (MDA-MB-231, MCF-7, U251, and MO59K). The genotoxic effect of AgNPs was demonstrated by an increase in the accumulation of γ-H2AX, which is considered a marker of DNA damage, and damage to telomeres. The alteration of telomeres after treatment with AgNPs was accompanied by a reduction in intracellular levels of TRF2, which plays a key role in protecting telomeres [137] and the downregulation of hTERT, the catalytic component of the enzyme telomerase. The expression of JNK is positively regulated by the phosphorylation of DNA-PKcs (DNA-dependent protein kinases), which are essential for the repair of DNA double-strand breaks [138]. Using a DNA-PKcs inhibitor reduced the phosphorylation and expression of DNA-PKcs, the levels of phosphorylated and total JNK, and the overall survival of tumor cells. The results of this study suggest that the inhibition of DNA repair mechanisms such as DNA-dependent protein kinases may increase the antiproliferative and cytotoxic effect of AgNPs against tumor cells [139].

**Table 1 life-13-00466-t001:** Results of studies pointing to the influence of AgNPs on cell cycle regulation and apoptotic/nonapoptotic cell death induction.

Capping	Size	IC_50_ or Used Concentration	Tumor Model	Cell Line	Mechanism of Action	Reference
*Annona muricata* aqueous leaf extract	80 ± 6.3 nm	6 µg/mL	Human lung cancer	A549	Induction of apoptosis ↓ Bcl-2, cyclin B, E, D ↑ Bax, p21, p53, caspases −3, −8, −9, Fas-L	[61]
Glucose	30 ± 5 nm	13.5 μg/mL	Human cervix carcinoma	HeLa	Induction of apoptosis S and G2/M cell cycle arrest	[140]
Polysaccharide	11 ± 5 nm	5.05–75 μg/mL	Human breast cancer	SKBR3, 8701-BC	Inhibition of colony formation and migration, ROS generation ↓ MMP-2, MMP-9, Akt, phospho-Akt, HSP70 ↑ Beclin-1, ATG5, ATG7, LC3A/B, p62	[120]
*Bergenia ligulata* aqueous extract		5–10 μg/mL	Human breast cancer	MCF-7	Induction of apoptosis, G2/M cell cycle arrest, ROS generation, inhibition of colony formation and migration ↓ MMP, Bcl-2 ↑ p53, phospho-p53, Bax, caspase −3	[59]
*Annona muricata* leaf extract	23 ± 14 nm	5–25 μg/mL	Human lung cancer	A549, Calu-1, BEAS-2B	G1 and G2 cell cycle arrest, mitochondrial ROS generation, and protein oxidation	[141]
Aqueous extract	63.1 ± 8.3 nm	150 μg/mL	Human breast cancer	MCF-7, T47D, MDA-MB-231	Induction of apoptosis, G0/G1 and S cell cycle arrest ↓ XLOC_006390, SOX4 ↑ miR-338-3p	[142]
*Carica papaya* leaf extract	10–20 nm	2.5/5 μg/mL	Human prostate cancer	DU145	Induction of apoptosis, G1 and G2/M cell cycle arrest, ROS generation ↓cyclin D1 ↑ p21, p27, Bax, cleaved PARP, and cleaved caspase −3	[62]
*Phlomis armeniaca* aqueous extract		10 µM	Human breast cancer	MCF-7, MDA-MB-231	DNA fragmentation, ROS production ↑ endonuclease G	[143]
*Macrotyloma uniflorum* seed extracts	91.8 nm	20–50 μM	Human ovarian cancer	PA-1	Induction of apoptosis, G2/M cell cycle arrest, mitochondrial membrane damage, and ROS generation ↑ caspase −3	[144]
Glucose	61 nm	860/1528 μg/mL	Human prostate cancer	DU145, PC-3	Induction of apoptosis, S phase cell cycle arrest, mitochondrial membrane damage, and ROS production	[92]
*Moringa oleifera* leaf extract	10–100 nm	7.5 µg/mL	Human leukemia	Kasumi-1	Induction of apoptosis, S phase cell cycle arrest ↑ BID	[145]
*Juniperus chinensis* extract	12.96 ± 5.65 nm	9.87 µg/mL	Human lung cancer	A549	Induction of apoptosis, ROS production, inhibition of migration and invasion, chromatin condensation, DNA, and nuclear fragmentation↓cyclin D1, MMP-2, MMP-9 ↑ p53, and caspase −3, −9	[29]
*Parmelia sulcata* extract	16 nm	30 µg/mL	Human breast cancer	MCF-7	Induction of apoptosis, ROS production↓TNF-α, IL-6, PCNA, cyclin D1, Bcl-2, and MMP ↑ Bax	[60]
*Artemisia arborescens* extract	4–30 nm	7 µg/mL	Human and cervical cancer	MCF-7, HeLa	Inhibition of colony formation and induction of apoptosis, cell cycle arrest in G1 phase	[146]
*Catharanthus roseus* aquoeus extract	1–100 nm	3.871 ± 0.18 µg/mL	Human hepatocellular carcinoma	HepG2	Induction of apoptosis, DNA damage/fragmentation, ROS and nitrite generation, and G2/M cell cycle arrest ↓ MMP	[81]
*Mentha arvensis* leaf extract	3–9 nm	1.56 μg/mL	Human breast cancer	MCF-7, MDA-MB-231	Induction of apoptosis, nuclear fragmentation↓Bcl-2 ↑ cleaved PARP, cleaved caspase −3, −9, and Bax	[147]
Naringeninaquoeus extract	6 nm	3 µg/mL	Human colorectal carcinoma	HCT116	Induction of apoptosis, ROS generation and lipid peroxidation, and DNA fragmentation ↓MMP, ATP, CCNB1, CCNB2 ↑ CYP1A1, CYP1B1, and GADD45G	[89]
*Matricaria chamomilla* aquoeus extract	45.21 nm	10–63 µg/mL	Human lung cancer	A549	S phase cycle arrest ↓ Bcl-2↑ Bax, caspase −3, −7	[148]
Aquoeus solution	2.6/18 nm	3–26 µg/mL	Human pancreatic ductal adenocarcinoma	PANC-1	Induction of apoptosis, ROS generation ↓ MMP, SOD1, SOD2, GPX-4, CAT ↑ nNOS, iNOS, eNOS, NO, NO_2_, and SOD3	[149]
*Artemisia oliveriana* extract	10.63 nm	3.6 µg/mL	Human lung cancer	A549	Induction of apoptosis, DNA fragmentation ↓Bcl-2 ↑ cleaved caspase −3, −9, Bax, and miR-192	[150]
*Nepeta deflersiana* extract	33 nm	10–50 µg/mL	Human cervical cancer	HeLa	Induction of apoptosis, ROS generation, and lipid peroxidation ↓MMP, GSH	[151]
*Manilkara zapota (L.)* leaf extracts	24 nm	8–25 µg/mL	Human colorectal carcinoma	HCT116	Inhibition of colony formation and migration, induction of apoptosis, and ROS generation ↓MMP, Bcl-2 ↑ cleaved caspase −3, −8, −9, PUMA, Bax, and cleaved PARP	[86]
Quinacrine-Based NPs	50–100 nm	0.6–1 µg/mL	Human squamous carcinoma	SCC-9	DNA damage, induction of apoptosis, and S phase cell cycle arrest↓Bcl-xL, cyclin E1, B1, A2, Cdc-25A, Cdc-2, Chk-1, Topo-IIα, RFC-1, Helicase, MRE11, RPA, Fen-1, XRCC-1, Pol-β, Pol-ε, DNA PKcs, NfкB ↑ Bax, p53, p21, γH2AX, and Chk-2	[63]
*Gossypium hirsutum* leaf extract	23.5–163.7 nm	40 μg/mL	Human lung cancer	A549	Induction of apoptosis, G2/M phase cell cycle arrest ↓ MMP ↑ caspase −3, −9, cytochrome *c*, p53, and Bax	[152]
*Zizyphus mauritiana* fruit extract	16 nm	28 μg/mL	Human breast cancer	MCF-7	Inhibition of colony formation, ROS generation ↓ total PARP ↑ caspase −8, FADD, and FAS	[75]
Root extracts of *Beta vulgaris L.*	30 nm	20–40 μg/mL	Human hepatoma cells	HuH-7cells	ROS generation, lipid peroxidation, chromosome condensation, induction of apoptosis, and necrosis ↓ GSH, MMP, Bcl-2 ↑ caspase −3, and Bax	[80]
Aqueous extract of *Coptis chinensis*	6–45 nm	5–20 µg/mL	Human lung cancer	A549	Inhibition of migration and invasion, DNA fragmentation ↓ Bcl-2, Bcl-xL↑ Bax, caspase, and cleaved caspase −3	[87]
*Geodorum densiflorum* rhizome extracts	25 nm	21.5/28 µg/mL	Human breast cancer, glioblastoma stem cells	MCF-7, GSCsk	Induction of apoptosis ↑ NFκB, TNFα, p21, TLR9, caspase −3, −8, −9, MAPK, JNK, p53, and FAS	[132]
*Rubus fairholmianus* extract	30–150 nm	2.5–10 µg/mL	Human breast cancer	MCF-7	Induction of apoptosis, DNA damage, ROS production ↓ MMP↑cytochrome *c* release, activity od caspases −3/−7, and Bax	[79]
*Ginkgo biloba* leaves aqueous extract	40.2 ± 1.2 nm	3/6 µg/mL	Human cervical cancer	HeLa, SiHa	Inhibition of proliferation and induction of apoptosis, inhibition of colony formation ↓ SOD, GSH-Px, mitochondrial cytochrome *c,* Bcl-2 ↑ cytosolic cytochrome *c,* Bax, and cleaved caspase −3, −9	[93]
AgNPs coated with polyvinylpyrrolidone (PVP)	15 nm	20–160 µg/mL	Human hepatocellular carcinoma	HepG2	ROS production↓ Bcl-2, MMP, NFκB ↑ cytochrome *c,* Bax, and cleaved caspase −3, −8, −9	[95]
*Albizia adianthifolia* aqueous leaf extract		43 μg/mL	Human lung cancer	A549	Induction of apoptosis and necrosis, DNA fragmentation, lipid peroxidation, and mitochondrial membrane damage ↓ ATP, GSH, ↑ p53, Bax, PARP-1, smac/DIABLO, and caspase −3/7, −8, −9	[94]
*Rhynchosia suaveolens* aqueous leaf extract	10–30 nm	4.2 μg/mL	Human ovariancarcinoma	SKOV3	ROS production, lipid peroxidation, inhibition of migration, and DNA fragmentation ↓ GSH ↑ p53, Bax, and caspase −3/7, −8, −9,	[96]
Polyvinylpyrrolidone	25 nm	37.5 µg/mL	Human breast cancer	MDA-MB-231	Induction of apoptosis, S phase cell cycle arrest, and DNA damage ↓ NADPH/NADP+, GSH/GSSG ↑ GRP78, PERK, phospho-eIF2α/total eIF2α, CHOP, and γH2AX	[110]
Ag-TF@^P^DOXtheranostic nanocomposite	185.9 nm	50 μg/mL (DOX)	Human breast cancer	MCF-7/ADR	Induction of endoplasmic reticulum stress ↓ P-gp ↑ cytoplasmic Ca^2+^ level, GRP78, PERK, CHOP, ATF4, and caspase −12	[109]
		0.2–125 µg/mL	Human colorectal carcinoma	HCT116	ROS generation, induction of mitochondrial dysfunction, ER stress, and apoptosis ↓ Mcl-1 ↑ phospho-ASK1, phospho-JNK1, Bim, Bad, and cleaved PARP,	[153]
β-cyclodextrin	9 nm	25 µM	Human breast cancer	MCF-7	Induction of mitochondrial fragmentation, mitochondrial ROS production, and apoptosis ↓ MMP ↑ phospho-DRP1, TOM 20, Bip, IRE-1α, and calnexin	[111]
Polyvinylpyrrolidone	30 nm	12.5 μg/mL	Human neuroblastoma	SH-SY5Y	Induction of apoptosis, mitochondrial dysfunction ↓ ATP, phospho-IP_3_R ↑ GRP78, phospho-PERK, phospho-eIF2α, ATF4, phospho-IRE-1, spliced XBP1, CHOP, calpain, caspase −12, PTEN, Bax, cytochrome *c*, and caspase −3, −9	[21]
Polyvinylpyrrolidone	67–78 nm	2–6 μg/mL	Human prostate cancer	PC-3	Induction of lysosomal injury, activation of protective autophagy ↓ATP, phospho-mTOR, mTOR, LAMP1, cathepsin D, phospho-PS6K, PS6K, LC3-I ↑ ATG3, ATG5, ATG7, ATG12, Beclin-1, phospho-AMPK, phospho-ACC1, ACC1, and LC3-II	[119]
*Klebsiella oxytoca* under aerobic conditions	11 ± 5 nm	1.25–10 μg/mL	Human breast adenocarcinoma	SKBR3	ROS generation, inhibition of colony formation, autophagolysosomes formation, and induction of apoptosis and autophagy ↓ Akt, phospho-Akt, HSP90, p62, MMP-2, MMP -9 ↑ ATG5, ATG7, Beclin-1, and LC3-I/II	[120]
Gallic acid	86.6, 38.13, and 59 nm	33.45 μg/mL	Human colorectal adenocarcinoma	HT-29	Induction of mitochondria-dependent apoptosis and late noncanonical autophagy ↓ ATG3, ATG12 ↑ cytochrome *c*, p53, Bax, caspase −3, −8, −9, −12, XBP1, CHOP, Beclin-1, and LC3-II	[123]
Polydopamine (PDA)-coated Au–AgNPs	96.7 ± 6.1 nm	20–80 μg/mL	Human papillary thyroid cancer	TPC-1	S phase cell cycle arrest and autophagosomes and autolysosomes formation ↓ DHODH ↑ p53, LC3-A/B	[154]
*Annona muricata* Extract	6–31 nm	17.34 µM	Human acute monocytic leukemia, human breast cancer	THP-1, AMJ-13	Induction of apoptosis and inhibition of colony formation ↓ MMP ↑ p53, LC3-I/II	[155]
*Brassica rapa* var. *japonica* leaf extract	15–30 nm	1–10 μg/mL	Human colorectal adenocarcinoma	Caco-2	Induction of apoptosis and necrosis, ROS generation, and DNA fragmentation ↓ GSH, NF-κB, Akt, mTOR↑ IκB α, p53, LC3-II, and caspase −3	[124]
	2.6/18 nm	0.5–2.5/10–50 μg/mL	Human pancreas ductal adenocarcinoma	PANC-1	Induction of apoptosis, autophagy, necroptosis, and mitotic catastrophe ↓ Bcl-2 ↑ Bax, MLKL, p53, RIP-1, RIP-3, and LC3-II	[125]
	7.9 nm	10 ppm	Pancreatic ductal adenocarcinoma	BxPc-3, AsPC-1	ROS generation, inhibition of colony and spheroid formation and migration, induction of paraptosis-like cell death, and induction of MAPK signaling ↓ RAD51 ↑ p62, LC3-I/II, caspase −3, TRAIL, phospho-MLKL, p42/44 (Erk1/2), GADD45, and phospho-MBP	[127]
Sinigrin	20 nm	1 μM	Human cervical cancer	HeLa	ROS generation and induction of apoptosis ↓ GSH, SOD, CAT, GPx, MMP, Bcl-2, Bcl-xL, Akt, Raf, MEK, Erk1/2, cyclin D↑ MDA, caspases −3, −6, −9, p21, p53, cytochrome *c*, Bid, Bax, Bak, JNK, p38, and NFκB	[23]
RGD peptide-conjugated Ag@Se@RGD NPs	72 nm	20 μg/mL	Human glioma	U251	ROS generation and induction of apoptosis ↑ phospho-JNK, phospho-p38, and phospho-Erk1/2	[133]
Polydopamine-coated Au–Ag NPs	200 nm	20–40 µg/mL	Human bladder cancer	T24	Induction of apoptosis and autophagy, S phase cell cycle arrest, ROS generation ↓ cyclin A, Bcl-2, Bcl-xL, procaspase −3, MMP, phospho-Erk1/2, phospho-Akt ↑ p21, caspase −3, −8, Bax, cytochrome *c*, and LC3-II,	[135]
β-sitosterol	5–55 nm	7/14 ng	Human hepatocellular carcinoma	HepG2	ROS generation, induction of apoptosis, ↓ mitochondrial cytochrome *c*, Bcl-2, ↑ Nfr2, cytoplasmic cytochrome *c*, Bax, caspase −3, −9, Apaf1, and p53	[156]

#### 2.2.4. AgNPs as ROS Inducers

It is well known that silver nanoparticles may trigger reactive oxygen species (ROS) production in mammalian cells. The quantity of produced ROS is greatly dependent on the kind of cell, its metabolic activity (i.e., the number of mitochondria), the redox state of the surrounding environment, and a host of other factors.

Their toxicity is dependent on the production of reactive oxygen species (ROS) [94,157,158,159,160], both directly (by electron donation to molecular oxygen, forming O2-) and indirectly (by interfering with mitochondrial structure and function, resulting in O2- leakage from the electron transport chain [161]. ROS consist of the superoxide anion (O2-), hydrogen peroxide (H_2_O_2_), and the hydroxyl radical (HO•). Under physiological settings, the reservoir of antioxidants in the cell keeps these signaling molecules at low concentrations [151,162]. However, excessive ROS production is linked to oxidative stress. The rise in ROS levels generated by exposure to AgNPs produces cytotoxicity, lowering cell proliferation rates, triggering macromolecule and organelle damage, and finally resulting in cell death. Therefore, the bulk of research suggests that nanosilver treatment increases ROS and oxidative stress. However, this has been contested by studies in which there was no rise in ROS [163]. This might be the result of various experimental circumstances or difficulties in identifying the real amounts of ROS using H2DCFDA, the primary dye used to detect ROS.

In the work of Piao et al., 2011 [164], oxidative stress was evaluated by measuring ROS generation (reported as DCF level) and lipid peroxidation level; the findings demonstrated concentration-dependent oxidative stress. Similar to the action of DOX on HepG2 cells, ingestion of the antioxidant glutathione mitigates the damage caused by kaempferol-coated AgNPs. After exposure to AgNPs, increased ROS generation and reduced GSH levels have been found in human Chang hepatic cells. The size of AgNPs has a significant effect on their method of action, since it influences their absorption by cells. It has been claimed that smaller NPs with their greater surface area may disperse and release ions more readily [165]. While coated 200 nm AgNPs could be taken up by cells to a greater extent through endocytosis, they induce distinct molecular pathways of cellular response [166]. Consequently, examining intracellular ROS production is a crucial sign of NPs-induced toxicity and might be regarded the first step in the toxicity cascades [167]. Due to the depletion of antioxidant (GSH) capability, cells undergo programmed cell death (apoptosis) in the event of excessive ROS production caused by the influence of NPs on cells, as seen in the work of Li et al., 2021 [168].

#### 2.2.5. AgNPs and MDR (Multidrug Resistance) Induced by Drug Efflux

Pharmacotherapy is an integral part of the treatment of most tumors. However, the occurrence of resistance to existing anticancer drugs remains a serious obstacle leading to insufficient therapeutic efficacy, which is still discussed in direct relation to efflux membrane transporters from the ATP-binding cassette transporters superfamily (ABC transporters) [169,170].

In humans, important efflux transporters are typically present in biological barriers, such as the intestine, liver, kidney tubules, blood–brain barrier, testis, and placenta, as well as in the membranes of cancer cells [171]. It was shown that such transporters expel xenobiotics, including drugs, or endogenous substances out of the body and out of the tumor cells [172]. Thus, they influence the disposition of various drugs at their molecular targets resulting in drug resistance. In addition, ATP-dependent efflux transporters may be involved in several other processes. For instance, they may influence apoptosis [173], inflammation [174] or tumor immunity [175].

Previous research has documented that anticancer resistance is mainly the result of the action of ATP-binding cassette subfamily B member 1 (ABCB1) protein, also known as P-glycoprotein (P-GP) or multidrug resistance protein 1 (MDR1). This protein is also responsible for the phenomenon of multidrug resistance [176]. Approximately 50% of human malignancies express ABCB1 at levels sufficient to lead to the development of chemoresistance [177]. According to the reviews, various anticancer drugs are substrates of ABCB1 [178]. Moreover, the chemotherapeutics from nonsubstrates of ABCB1 may increase its mRNA expression [179]. Complementary to this, some tumor cells may also express other efflux transporters responsible for drug resistance, such as ABCG2 (breast cancer resistance protein, BCRP) [180] or ABCC1 (multidrug-resistance-associated protein 1, MRP1) [181]. That is why looking for new drugs with the potential to overcome drug efflux is an intensive challenge for experts from various fields. Attention is also given to suitable inhibitors of efflux membrane pumps.

Several recent studies demonstrated that silver nanoparticles might serve as potential modulators of ABC transporters and promising chemosensitizing agents, as summarized in Table 2 and described below.

Earlier experiments showed a notable antiproliferative action of silver nanoparticles or their analogues modified with TAT-enhanced cell-penetrating peptide (AgNPs-TAT), even in drug-resistant cells, such as breast cancer adenocarcinoma cell line resistant to doxorubicin (MCF-7/ADR) or drug-resistant murine melanoma cells (B16) [188]. Additionally, these compounds also exhibited antitumor effects *in vivo*, which were confirmed in mice bearing resistant skin melanoma. Generally, AgNPs-TAT analogues were more efficient in the inhibition of tumor volume, probably due to their improved cellular uptake.

Next, metal-containing nanoparticles of a certain specific size also exerted an increased nuclear uptake by bypassing the membrane ABCB1 transporter, which allows traditional cytostatics to more effectively reach their drug targets in the nucleus. Such a property of nanoparticles was demonstrated in a study focused on the preparation of a silver–gold nanorod, which was used as a specific size photocotrollable nuclear-uptake nanodrug delivery system (nanotruck) for cytostatics such as doxorubicin [189].

Later, decreased efflux activity after exposition with AgNPs 23 nm was determined in the Madin–Darby canine kidney type II cells transfected with human ABCB1 (MDCKII–MDR1) [184]. Moreover, in vitro analyses confirmed inhibitory action on efflux activity for Ag ions (AgNO_3_) but surprisingly not for other AgNPs (20 and 27 nm) and larger 200 nm Ag particles. Both Ag ions and some AgNPs (23 and 27 nm) also exerted an increase in calcein fluorescence in vivo (model using *Daphnia magna juveniles*). However, the effects of AgNPs in vivo experiments were neither related to a change in ABCB1 gene expression, nor to an effect on cell membrane permeability unrelated to efflux proteins (no significant inhibition of efflux in wild MDCKII cell line). The differences in the actions of individual AgNPs were explained by the different origin of those compounds, which could lead to differences in their stability, the amount of released Ag ions or differences in their uptake by cells.

The interference of AgNPs with the ABCB1 transporter was also recorded in the work of Kovács et al. [183]. The authors reported the antiproliferative effects of synthesized silver nanoparticles in both parental drug-sensitive (Colo 205) and ABCB1 overexpressing human colon adenocarcinoma cells (Colo 320). Imaging techniques revealed that AgNPs might be attached to the surface of (scanning electron microscopy, SEM) or taken up (transmission electron microscope, TEM) by cancer cells of both tested cell lines. Next, a decreased efflux activity of ABCB1 transporter was observed after treating with the tested compound in resistant Colo 320 cells. It was shown that it might be the result of the decreased ABCB1 expression on mRNA and protein levels. The experiments also brought an important discovery that silver nanoparticles were found to have synergistic interactions with several conventional cytostatics considered ABCB1 substrates (methotrexate, cisplatin, carmustine, bleomycin, and vinblastine) and verapamil (a known ABCB1 inhibitor).

In the same line, other researchers prepared a combination of camptothecin nanocrystals with silver (CPT-Ag nanocrystals), which inhibited cell viability in breast cancer cell line (MDAMB231), naturally resistant to crude camptothecin [186]. Moreover, this combined nanocompound exerted higher cellular uptake in several cancer cell lines (human non-small-cell lung carcinoma, A549; human cervical adenocarcinoma, Hela; and human breast adenocarcinoma cell lines, MCF-7, MDA-MB-231, and SKBR3) compared with crude CPT nanocrystals and diminished the efflux of rhodamine (MDA-MB-231). Similarly, paclitaxel nanocrystals combined with silver nanoparticles were more efficiently taken up by cancer cells [190].

Equally important, AgNPs may interact not only with useful anticancer drugs but also with other ABC modulators originating from the environment. It was presented as increased rhodamine 123 accumulation when combinations of AgNPs with heavy metals (cadmium, Cd, mercury, and Hg) were used in toxicological experiments performed on human hepatocellular carcinoma cells (HepG2 cell line) [185]. Next, co-administration of AgNPs with Cd exerted higher inhibitory effects on ABC transport than combination with Hg. The nanoparticles also increased the toxicity of heavy metals, depending on the type of toxic effect observed and the concentration of the particular metals. Free AgNPs were able to significantly inhibit the efflux activity only at higher concentrations (3.5 µg/mL vs. 0.35 µg/mL) after 24 h of exposure.

Other data verified that larger AgNPs (75 nm vs. 5 nm) are more efficient in the inhibition of ABCB1 efflux activity in a doxorubicin-resistant human breast cancer cell line (MCF-7/KCR), although nanoparticles of both diameters were taken up by cancer cells [112]. Surprisingly, this action was not related to the ABCB1 protein level in treating cells. The researchers suggest that it might be probably due to decreased plasma fraction of ABCB1 transporter induced by critical depletion of calcium in endoplasmic reticulum and increased endoplasmic reticulum stress. Another essential point is that 75 nm-sized AgNPs sensitized resistant cancer cells to doxorubicin-induced apoptosis.

Furthermore, the effects of AgNPs on MDR phenotype depend not only on their molecular size, but also on the type of affected tumor cells. Despite Western blotting analysis confirming the presence of ABCB1 transporter in several cell lines of different tissue origin (human non-small-cell lung carcinoma, A549; human hepatocellular carcinoma, HepG2; and human colon adenocarcinoma, SW620), AgNPs modestly diminished ABCB1 protein expression only in A549 and SW620 cells [182]. Overall, the lowest protein expression of ABCB1 protein was noted in lung cancer cells. Gene expression assessment of treated cells determined diminished expression of most ABC transporters (not ABCB1) mainly after 12 h of incubation in A549 and HepG2 cell lines, while gene expression in SW620 cells showed the opposite direction. It has been suggested that the gene transcription of some ABC transporters might be under the control of nuclear factor kappa B, which is regulated by oxidative stress induced by incubation with AgNPs. Subsequently, AgNPs were able to inhibit the efflux activity of ABCB1 shown by a significantly increased calcein accumulation observed in two cell lines (HepG2 and SW620). On the other hand, no such inhibitory activity of AgNPs on the ABCC1 transporter was found, despite their impact on corresponding gene transcription.

Recently, specific nanoscale crystals in the form of quantum dots (QDs) were developed as a novel potential drug delivery system. This technique was also applied for preparing QD nanocrystals containing silver nitrate with other components (indium chloride, zinc stearate, and sulfur), which were modified by adding three different drug carriers: 11-mercaptoundecanoic acid, L-cysteine, and lipoic acid, all of them with bound folic acid and doxorubicin (QD–MUA–FA–DOX, QD–Cys–FA–DOX, and QD–LA–FA–DOX, respectively) [187]. In this research, QD nanocrystals were made in order to improve the entry of a chemotherapeutic agent (doxorubicin) into lung cancer cells (A549 cell line) by bypassing drug efflux. As explained in the literature, adding FA is especially responsible for improved cellular uptake of substrates of ABCB1, such as doxorubicin (Dzwonek et al., 2018) [191]. Similarly, Ruzycka-Ayoush et al. [187] confirmed increased cytotoxic effects of doxorubicin for all QDs carrying this chemotherapeutic with FA. Moreover, these nanoconjugates exerted more selective effects on tested cancer cells than on noncancerous fibroblast cells (NIH/3T3). It should be mentioned that inhibition of efflux might increase the toxicity of ABC substrates in healthy cells. Thus, selectivity of novel ABC inhibitors is an important requirement.

Overall, the data obtained from the literature indicate the need for further and deeper study of the interference of AgNPs with ABC proteins, as well as the need for a more detailed clarification of the mechanisms and regulatory pathways responsible for the emergence of the observed effects.

#### 2.2.6. AgNPs and Energy Metabolism

The metabolism of tumoral cells differs significantly from that of healthy cells; whereas oxidative phosphorylation is the primary energy-generating pathway in healthy cells, cancers are adapted for rapid growth in hypoxic and acidic environments, so glycolysis is the preferred pathway to synthesize ATP [192,193].

Lee et al. showed that exposure to 5 nm AgNPs decreased glucose intake and lactate generation in HepG2 cells [194], while Miranda et al. identified a downregulation of major glycolysis enzymes in hepatoma cell lines treated to 10 nm AgNPs [195]. In addition, AgNPs are known for their deleterious effect on mitochondria and the mitochondrial respiratory chain, which not only causes electron leakage and O2- production but also triggers apoptosis pathways (via cytochrome c release and caspase activation), inhibiting another essential ATP-generating pathway [79,160,196]. These characteristics may be advantageous for AgNPs to inhibit the bioenergetics of cancer cells and induce cell death.

#### 2.2.7. Tumor Invasiveness and Metastasis as a Target for AgNPs

Metastasis is a complicated process with two distinct phases. In the first phase, cancer cells move from the main tumor into the surrounding tissues and vascular networks. Cells leave the circulation, invade new tissues, and create additional tumors in the second stage. Cancer cells need epithelial–mesenchymal transition (EMT), a biological process that changes immobile epithelial cells into motile mesenchymal cells by breaking cell adhesion, for their invasive and migratory capacities. A transition in gene expression from epithelial to mesenchymal phenotype is caused by transcriptional regulation using Snai1, Snai2, Snai3, Zeb1, and Zeb2 transcription factors that govern the expression of EMT genes, such as Cdh1 or Pir [197,198].

Numerous investigations have established that AgNPs inhibit tumor cell migration and invasion in a concentration- and dose-dependent manner. Migration and invasion are significant characteristics of the development and worsening of cancer [61,197,198,199]. Although it has been discovered that AgNPs suppress tumor invasion [200], the underlying mechanism remains unknown. It is speculated that AgNPs may inhibit the protein production of cytokines and growth factors in cancer cells, as well as the enzymatic activity of MMPs.

Authors Que et al., 2019 [201] describe that AgNPs have the potential to suppress the migration and invasion of lung adenocarcinoma cells *in vitro*. They suggest that the size of nanoparticles plays a significant role, with 13 nm AgNPs being the most effective. The effect starts decreasing with the size of 45 nm and completely vanishes for 92 nm AgNPs.

The findings of the authors Sathishkumar et al., 2016 [202] clearly show that AgNPs may block cancer cell migration and lower the likelihood of breast cancer spread. In addition, the findings of the cell migration investigation indicate that AgNPs suppressed cancer cell migration in addition to inhibiting cancer cell growth via inducing apoptosis.

On MDA-MB 231 cell models, the toxicity, proliferation, and antimetastatic properties of FILE (*Ficus ingens* leaf extract) AgNPs were evaluated using the trypan blue, MTT, and wound healing assays, respectively. Metastasis of MDA-MB 231 cells was inhibited by FILE-AgNPs in a dose-dependent manner, with 10 µg/mL and 5 µg/mL inhibiting by 96% and 75%, respectively. The FILE-AgNPs generated are exceptional prospects for the treatment of cancer patients and other cancer-related situations [203].

The work of authors Mata et al. (2023) [204] identifies the therapeutic effects of biogenic *Abutilon indicum* silver and gold nanoparticles (AIAgNPs and AIAuNPs) in Wistar rats with colorectal cancer (CRC) caused by 1, 2-dimethyl hydrazine (DMH). The bioavailability of AIAgNPs and AIAuNPs in colon tumors of CRC rats treated with nanoparticles (NPs) was shown by the high localization of AIAgNPs and AIAuNPs, as measured by ICP-OES (inductively coupled plasma atomic emission spectroscopy). AIAgNPs and AIAuNPs dramatically increased antioxidant enzyme levels, such as catalase, SOD, GSH, and GPx, and decreased lipid peroxidation (LPO), compared with the conventional medication paclitaxel. Compared with paclitaxel, AIAgNPs and AIAuNPs demonstrated considerable protection against metastases in the histological analysis. Important CRC signaling components of the Wnt pathway, catenin and Tcf-4 levels, were considerably reduced in CRC rats treated with AIAgNPs and AIAuNPs compared with paclitaxel.

#### 2.2.8. AgNPs and Epigenetics

The growing usage of nanoparticles (NPs) in a variety of applications necessitates an accurate evaluation of their potential toxicity to people. Initially, research on the toxicity of NPs concentrated on cytotoxic and genotoxic effects, but more recently, epigenetic modifications generated by nanoparticles have received more attention.

Several forms of nanoparticles have been shown to alter miRNA expression [205,206,207,208,209,210,211] and DNA methylation [209], as well as histone modifications such as acetylation [209,210], methylation [211], and phosphorylation [212,213]. Silver (AgNPs), gold (AuNPs), and superparamagnetic iron oxide nanoparticles are the three most frequent nanomaterials employed in medicine and industry (SPIONs). It has been established that both AgNPs and AuNPs change miRNA expression and cause DNA methylation. Furthermore, it has been shown that both kinds of nanoparticles change the phosphorylation of histone H3.

So far, the influence of AgNPs on miRNA expression has only been examined *in vitro*, with substantial alterations in miRNA expression seen in human dermal fibroblasts [214], Jurkat cells [206], and brain cells [207].

The acetylation of the lysine tail of histones by histone acetyltransferases (HATs) allows for greater expression, while deacetylation by histone deacetylases (HDACs) condenses the DNA and renders it inaccessible for expression [215]. A549 cells treated with toxic nanosilver (PVP-coated; 21.74 nm) were reported to undergo deacetylation of H3 [209]].

#### 2.2.9. Epithelial–Mesenchymal Transition Inhibition by AgNPs

Epithelial–mesenchymal transition (EMT) has emerged as a critical regulator of cancer cell invasion and metastasis. In addition to acquiring migratory/invasive capabilities, the EMT process is intimately linked to the formation of cancer stem cells (CSCs), hence contributing to chemoresistance (Figure 4).

Despite the fact that EMT is a significant therapeutic target for cancer therapy, its clinical applicability is currently restricted for a number of reasons, including tumor-stage heterogeneity, molecular–cellular target specificity, and adequate drug delivery. Regarding this last issue, various nanomaterials may be employed to inhibit EMT induction, hence providing unique treatment strategies against a variety of malignancies. These nanosystems are used to enhance the biodistribution and accumulation of chemotherapeutic medicines at the target location, with promising outcomes in preclinical and clinical investigations [216].

Gallic acid-coated silver nanoparticles suppress EMT and sensitize cancer cells to radiation-induced metastasis; they were able to downregulate EMT markers such as vimentin, N-cadherin, and Snail, while increasing E-cadherin expression in non-small cell lung cancer (NSCLC) [217]. However, the application of AgNPs is restricted because of their cytotoxicity and potential to stimulate proliferation and metastasis, as seen in colon adenocarcinoma cancer [218].

In the experiments of [219], HA (hyaluronic acid)-conjugated silver nanoparticles with graphene quantum dots were also successfully and specifically targeted to pancreatic CSCs. The toxicity of silver nanoparticles was reduced by linking carboxymethyl inulin to the nanoparticles.

Original figure made for this review using the Canva software.

### 2.3. AgNPs and Angiogenesis

Angiogenesis, or so-called “neovascularization”, is the complicated process of the growth of new blood vessels from existing ones. It is considered a complicated process because it leads to the proliferation of endothelial cells and subsequently to their migration, permeability, and the final formation of new blood vessels [220]. Neovascularization is essential for physiological processes such as growth and wound healing. An imbalance between proangiogenic and antiangiogenic factors leads to pathological angiogenesis, which is associated with the acceleration of several diseases, including the growth of malignant cells, ocular problems (diabetic retinopathy), and inflammatory diseases. This process is essential in cancer spread and invasion and later in the process of metastasis, and it was first hypothesized by Folkman (1971) more than 50 years ago [221]. A 2 mm^3^ tumor cannot grow without blood supply, so it stimulates angiogenesis as an early characteristic of tumor growth [222,223]. Angiogenesis plays an irreplaceable role in the process of metastasis formation. When the blood flow is large enough, parts of the tumor are released into it and travel to other parts of the body. Stopping the formation of new blood vessels leads to the termination of the tumor growth process and the cessation of metastases. The mechanism is based on the fact that they will not be supplied by oxygen and nutrients from the blood; moreover, toxic substances will not be released from it. Interrupting the formation of blood vessels is one of the methods of therapy for various types of cancer, regardless of the site of occurrence [224]. Antitumor therapeutic strategies can be divided into five groups: (a) antiangiogenesis, (b) breaking the immunosuppressive tumor microenvironment, (c) vascular normalization, (d) vascular blockade, and (e) vascular disruption [225].

The first antiangiogenic drug used is the monoclonal antibody Bevacizumab (Avastin), which is used to treat colorectal cancer by targeting overexpressed vascular endothelial growth factor (VEGF) proteins [226].

Angiogenesis can be monitored at in vitro and in vivo levels. Different cell lines are used to determine the effect of nanoparticles on neovascularization, such as human umbilical vein endothelial cells (HUVEC) [227], human retinal microvascular endothelial cells (HRMEC), bovine retinal endothelial cells [228], and bioartificial renal epithelial cells (BREC) [229]. The amount and activity of vascular endothelial growth factor (VEGF) and fibroblast growth factor (FGF) as essential promoters of angiogenesis are also deeply studied [229,230,231]. Inhibition of angiogenesis has also been observed in some cancer cell lines such as breast MCF7 [227]. Various animal models can be used, for example, the *in ovo* or *ex ovo* chick embryo chorioallantic membrane (CAM) assay [231,232], rat aortic ring model [233], rabbit cornea, mouse Matrigel implant assay [231] or C57BL/6 mice [229].

Nanoparticles can play an irreplaceable role in antiangiogenic cancer therapy [234,235,236]. They can reduce the shortcomings of antiangiogenic therapy by reducing side effects, optimizing efficacy, and precisely delivering diverse drugs, so realizing the combined therapy [237]. They have the potential to be used for their antiangiogenic effects or as drug transporters. Nanoparticles exert their antiangiogenic effect by targeting different angiogenic pathways. The inhibitory effect of silver nanoparticles on angiogenesis prepared by various methods and also by green synthesis has been reported [227,229,232]. The use of plant material (e.g., *Salvia officinalis* [232] and *Achillea biebersteinii* [233]) or bacteria (*Bacillus licheniformis* [230]) is shown for the preparation of silver nanoparticles with demonstrable antiangiogenic effects. The big advantage of using AgNPs is based on the fact that they are cost-effective, biocompatible, and their preparation is environmentally friendly [230].

#### 2.3.1. In Vitro Mechanism

Endothelial cells (ECs) are one of the targets during angiogenic therapy. It was observed that AgNPs changed the morphology of ECs (absence of cell–cell contact, normotonic cell shrinkage, and presence of apoptotic bodies), avoiding their proliferation, which led to a reduction in sprout and tube formations [228,230,238]. It can be concluded that the antiangiogenic effect of AgNPs is based on their direct interaction with ECs. HUVEC migration assays are provided for the detection of angiogenesis, since endothelial cell migration is an essential process for the formation of new blood vessels [227]. Another observed mechanism of AgNPs to neovascularization is based on the interaction with VEGF. AgNPs stopped VEGF-induced cell proliferation [238], migration [229], and vascular permeability [228], which led to the inhibition of angiogenesis. Angiogenesis requires various growth factors and cytokines (VEGF, FGF-2, TNF-α, IL-1, TGF-β, PDGF, MMP, and NO). AgNPs have a negative effect on genes such as VEGF-A and block signaling pathways such as the HIF signaling pathway [227], Src kinase pathway [228], and PI3K/Akt pathway [229].

AgNPs 10 nm in size have adverse effect on placental formation on in vitro models. The suppression of the expression of placental angiogenesis marker s-Flt-1 e15a was observed in forskolin-treated BeWo cells. These data suggest that silver nanoparticles could suppress formation of the syncytiotrophoblast and induce placental dysfunction and the following poor pregnancy outcomes [239].

#### 2.3.2. In Vivo Mechanism

AgNPs tested in the CAM assay have been shown to cause obstruction in the microcirculation [240,241]. After AgNPs treatment, distinct morphological changes indicating unhealthy organization were observed in the ring test of the rat aorta. Specifically, it involves minimizing cell spreading patterns, and cytoskeleton dysfunction has been observed [233]. A similar result was also found in the Matrigel implant test on mice, which showed inhibition of the formation of new blood microvessels [228,229]. Microscopic examination of the effect of AgNPs on angiogenesis in all three in vivo models showed a reduction in angiogenesis, as expressed by changes in the structure of blood vessels.

#### 2.3.3. Proangiogenic Effects

In contrast to many studies determining the negative effect of AgNPs on angiogenesis, there is one publication that confirms the proangiogenic effect. The authors postulated that the effect is probably based on the size of NPs and their method of synthesis. They used AgNPs coated with PVP 2.3 nm in size, whereas the antiangiogenic effect was observed in the case of bigger NPs (10–150 nm, mainly 50 nm) prepared by biosynthesis. Their next interpretation is based on the fact that the antiangiogenic effect in the presented studies is supposedly cooperating with the cytotoxic effect of AgNPs in high concentrations or after prolonged exposure [242].

In Table 3, the effects of different nanoparticles on angiogenesis tested on various models are summarized.

### 2.4. Anticancer Effects of AgNPs Detected In Vivo

In a research study by Sangiliyandi et al. [247], female Swiss albino mice were injected with Dalton’s lymphoma ascites (DLA) to induce tumors. The next day, mice were injected intraperitoneally with 500 nM of 50 nm nanosilver, which was the EC50 for DLA cells, for a period of 15 days. Caspase-3 was activated, DNA was fragmented, and DLA cells underwent apoptosis, which was not unexpected at this treatment dose. In contrast, noncancerous mice treated with 500 nM nanosilver exhibited no toxicity symptoms such as loss of appetite, weight loss, weariness, or fur color change. The nanosilver therapy considerably aided the tumor-bearing mice. They had much fewer malignant DLA cells in their peritoneal fluid and lived 50% longer than tumor-bearing mice that had not been treated. In addition, the nanosilver therapy reduced extra ascetic fluid by 65 percent, restoring normal body weight and white blood cell and platelet counts in the ascetic fluid. In this research, nanosilver therapy decreased cancer development in mice without creating any harmful consequences. Male Swiss albino mice with tumors developed from DLA cells were treated intraperitoneally with nanosilver (35 g/kg body weight) for 10 days after the injection of cancer cells, and this therapy was shown to be comparable to treatment with the anticancer medicine 5-Fluorouracil (20 micrograms per kilogram).

Treatment with nanosilver (citrate-coated, 5 nm) injected peritumorally around murine lung squamous tumor cells (KLN 205) in female immunologically competent DBA/2 mice and immunodeficient NOD SCID mice initially decreased the growth of the tumors in both mouse strains. Following this therapy, the effects of nanosilver progressively faded in the tumors of immune-deficient NOD SCID mice, and the tumor growth rate returned to its starting level. After treatment with nanosilver, the growth rate of tumors in immunocompetent DBA/2 mice was drastically decreased and did not recover, presumably owing to the immune response generated by the nanosilver therapy in these animals [248].

The levels of p53, p21, and cleaved caspase-3 rose in the liver tissue of male Sprague Dawley rats given orally with up to 100 mg/kg/day PVP-coated nanosilver (20–30 nm) for 90 days, then declined at higher doses, where autophagic cell death was believed to occur [209].

There is less research investigating AgNPs’ anticancer effect in animal models, however, these findings [248,249,250,251] indicate that AgNPs decrease tumor development *in vivo*. In vivo studies concentrate mostly on tumor development (tumor volume), but the influence of AgNP on the metastatic potential of tumors is seldom investigated (Table 4). For example, it has been documented that the gold-core silver-shell nanoparticles inhibit tumor growth and metastatic dissemination of 4T1 tumors in mice [250]. Nevertheless, Hu et al. [91] demonstrate that Angstrom-scale silver particles reduce both osteosarcoma development and its metastatic potential in nude mice with osteosarcoma. According to studies by Chakraborty et al. [251] and Manshian et al. [248], AgNPs’ anticancer efficacy in vivo is dependent on the activation and/or augmentation of the anticancer immune response.

## 3. Significant Challenges in the Clinical Application of AgNPs

Due to the complexity of silver nanoparticle cancer treatment, the first step is to comprehend and describe cancer. The heterogeneity of tumors necessitates the addition of receptor profiling and high-throughput personalized tumor analysis, such as DNA sequencing and transcriptomic, proteomic, and metabolomic analyses, to identify mutations, signaling pathways, and cellular characteristics that have been compromised during tumor development and progression. On the basis of these findings, customized and patient-specific multifunctional silver nanoparticles should be manufactured. These modifiable characteristics include the optimum size/shape and surface charges for effective pharmacokinetics, including the accumulation of tumor-specific AgNPs.

In addition, surface modifications for active cancer cell targeting, such as conjugating AgNPs with receptor ligands, antibodies, and cell-penetrating peptides, should be accounted for in the design of tailored nanoparticles. Lastly, the deployment of such complex AgNPs must be accompanied by treatment techniques, such as chemotherapy and radiation, that enhance each other synergistically. Prior to any of these applications, however, a comprehensive toxicological assessment must be conducted to ensure biocompatibility and a safe and effective AgNP-based oncotherapy. Despite the current obstacles, there is no question that metal nanoparticles will soon be included into authorized therapeutic treatment protocols. This nanoparticle-based medicine may soon become a reality. However, the dilemma of how to continue persists.

## 4. Conclusions and Future Perspectives

Significant basic and translational research has been conducted over the last decade to bring nanosized materials to cancer therapy [255]. As a consequence, various nanomaterial-based cancer therapy approaches have been tested in clinical studies [255] to date. Although they are mostly liposome-encapsulated chemotherapeutic medicines, other nanomaterials, particularly metal-based nanostructures, have lately emerged as effective therapeutic agents. Due to their well-recognized antimicrobial properties, silver nanoparticles (AgNPs) are the most commonly employed among them in a variety of therapeutic applications [256]. In addition to possessing antimicrobial capabilities, AgNPs also show unique cytotoxic effects against mammalian cells; these traits make silver-based nanoparticles potentially suitable in cancer treatment. A rapidly expanding body of scientific evidence supports their potential use as anticancer medicines.

Current science focuses on the biomedical properties of nanoparticles; however, there are concerns regarding their long-term toxicity. Recent trends in the production and surface manipulation of nanoparticles, such as the use of characteristic biogenic capping agents, have enabled the preparation of nontoxic, surface-functionalized, and monodispersed nanoparticles for medical applications. Therefore, the selection of suitable capping groups is the key in the stabilization of colloidal solutions and their absorption into living cells. Green capping agents with their roles in the synthesis of nanoparticles include, for example, proteins, carbohydrates, amino acids, lipids, extracts from fungi, algae, bacteria, and plants. Although designing capped nanoparticles exhibiting significantly improved biomedical and pharmaceutical properties compared with uncapped nanoparticles is very challenging, there is a growing body of evidence highlighting the beneficial role of capped agents in various biological applications [257,258,259].

Despite the promising findings of AgNPs as a novel treatment technique, they have not yet been put into clinical practice, mostly owing to a lack of information on their behavior and toxicity in people. Prior to its practical use, a comprehensive knowledge of the AgNP-induced effects on single cells, cancer tissues, and organs is required. To compare data from various labs and develop an agreement about their toxicity and pharmacokinetics, comprehensive nanoparticle characterization and standard experimental design are required. In addition, innovative targeting and biomimetic techniques, such as AgNPs coated with cancer cell membranes, should be investigated for the therapeutic use of AgNPs.

## Figures and Tables

**Figure 1 life-13-00466-f001:**
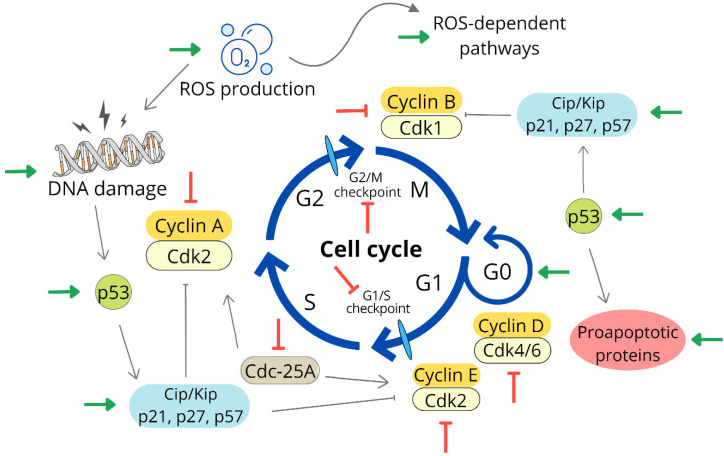
Cell cycle regulation modified by AgNPs. Green arrow—activation by AgNPs; red arrow—inhibition by AgNPs. Original figure made for this review using the Canva software.

**Figure 2 life-13-00466-f002:**
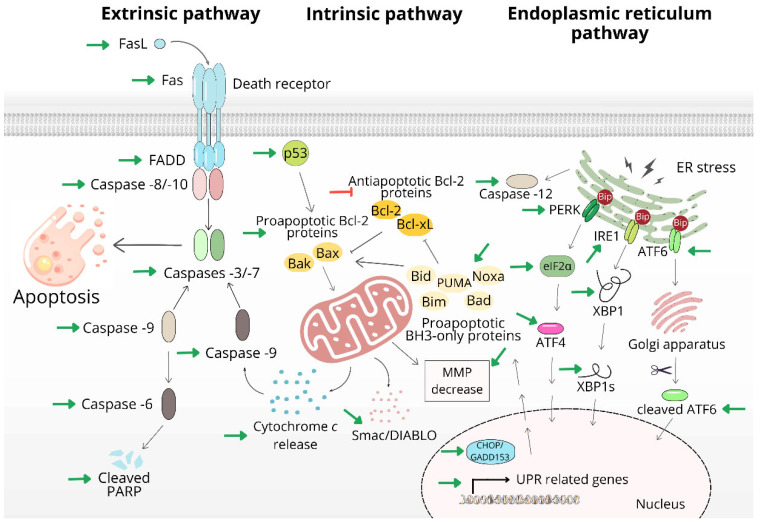
Molecular mechanisms of extrinsic, intrinsic and endoplasmic reticulum pathways involved in AgNPs-induced apoptosis. Green arrow—activation by AgNPs; red arrow—inhibition by AgNPs. Original figure made for this review using the Canva software.

**Figure 3 life-13-00466-f003:**
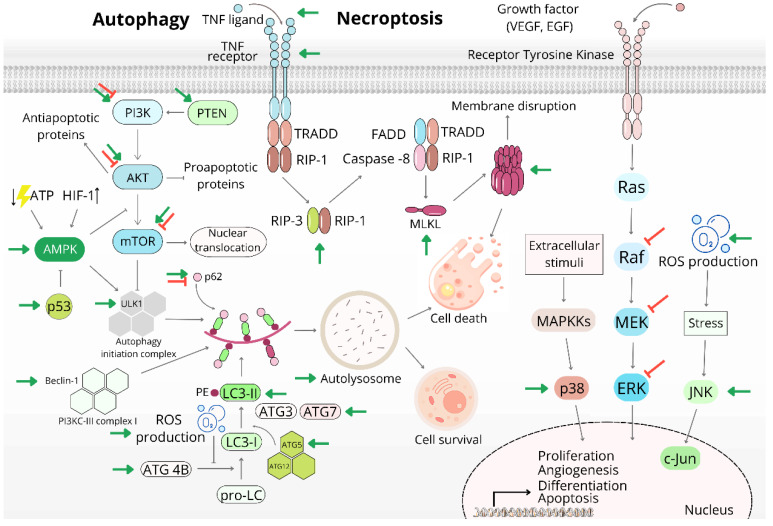
Autophagic, necroptotic and MAPK signaling pathways involved in the mechanism of action of AgNPs. Green arrow—activation by AgNPs; red arrow—inhibition by AgNPs. Original figure made for this review using the Canva software.

**Figure 4 life-13-00466-f004:**
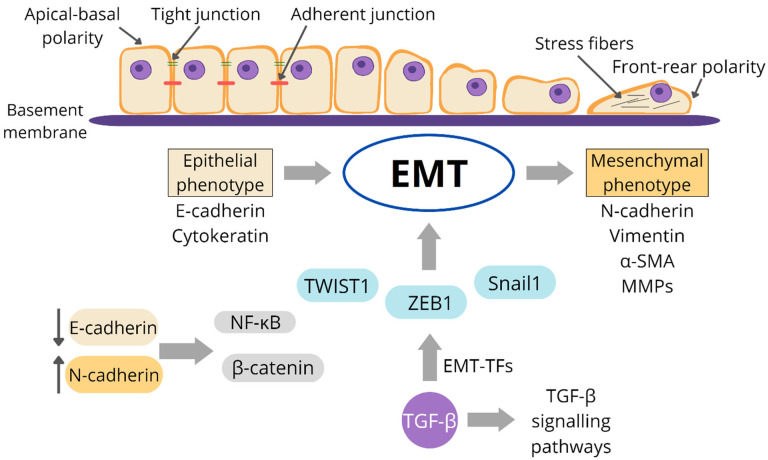
Molecular regulation of epithelial–mesenchymal transition (EMT).

**Table 2 life-13-00466-t002:** Effects of silver nanoparticles on ABC transporters.

Effects	Cell Line or Organism	Compound (Source, Particle Diameter)	Reference
**Decreased mRNA expression of most ABC transporters (not ABCB1)**	A549	Bare AgNPs (C, 20 nm)	[182]
	HepG2	Bare AgNPs (C, 20 nm)	[182]
**Increased mRNA expression of most ABC transporters (not ABCB1)**	SW620	Bare AgNPs (C, 20 nm)	[182]
**Decreased ABCB1 mRNA expression**	Colo 320	Citrate-coated AgNPs (S, 28 nm)	[183]
**Decreased ABCB1 protein expression**	Colo 320	Citrate-coated AgNPs (S, 28 nm)	[183]
	A549	Bare AgNPs (C, 20 nm)	[182]
	SW620	Bare AgNPs (C, 20 nm)	[182]
**Decreased ABCB1 efflux activity**	MDCKII–MDR1	AgNPs (S, 23 nm)	[184]
	Colo 320	Citrate-coated AgNPs (S, 28 nm)	[183]
	HepG2	AgNPs (S, 1–2 nm)	[185]
	HepG2	AgNPs (S, 1–2 nm) + Cd	[185]
	HepG2	AgNPs (S, 1–2 nm) + Hg	[185]
	A549	CPT-Ag NC (S, 300–900 nm)	[186]
	Hela	CPT-Ag NC (S, 300–900 nm)	[186]
	MCF7	CPT-Ag NC (S, 300–900 nm)	[186]
	MDAMB231	CPT/Ag NC (S, 300–900 nm)	[186]
	SKBR3	CPT/Ag NC (S, 300–900 nm)	[186]
	MCF-7/KCR	Citrate-coated AgNPs (S, 75 nm)	[112]
	HepG2	Bare AgNPs (C, 20 nm)	[182]
	SW620	Bare AgNPs (C, 20 nm)	[182]
	*Daphnia magna juveniles*	AgNPs (S, 23 nm and 27 nm)	[184]
**Chemosenzitizing effect**	Colo 320	Citrate-coated AgNPs (S, 28 nm)	[183]
	HepG2	AgNPs (S, 1–2 nm) + Cd	[185]
	HepG2	AgNPs (S, 1–2 nm) + Hg	[185]
	A549	CPT-Ag NC (S, 300–900 nm)	[186]
	Hela	CPT-Ag NC (S, 300–900 nm)	[186]
	MCF7	CPT-Ag NC (S, 300–900 nm)	[186]
	SKBR3	CPT-Ag NC (S, 300–900 nm)	[186]
	MDAMB231	CPT/Ag NC (S, 300–900 nm)	[186]
	MCF-7/KCR	Citrate-coated AgNPs (S, 75 nm)	[112]
	A549	Ag/In/Zn/S QDs-FA-DOX (S, 15.1–22.5 nm)	[187]

**Table 3 life-13-00466-t003:** The influence of AgNPs on angiogenesis detected on various models.

Type of Synthesis	Characterization of Nanoparticles			Angiogenesis		References
	Size		Shape			
	Range	Average		Methodology	Effects	
Antiangiogenic Effects						
Green synthesis using *Salvia officinalis*	1–40 nm	16.5 ± 1.2 nm	spherical, pentagonal	*in ovo* CAM	AgNPs reduced amount of total hemoglobin, the number, and the lengths of the vessels.	[232]
Green synthesis using *Achillea biebersteinii*	5–35 nm	12 ± 2 nm	hexagonal, pentagonal, and spherical	rat aortic ring model	AgNPs (200 μg/mL) reduce at 50% the length and number of vessel-like structures.	[233]
Green synthesis using *Decalepis hamiltonii*	heterogeneous in size		heterogeneous	Erlich Ascites murine carcinoma model	Significant changes in the neovasculature were observed in the treated mice; it means that AgNPs have the antiangiogenic effect.	[243]
Green synthesis using *Dictyota ciliolata*	n. s.	100 nm	spherical	*in ovo* CAM	AgNPs caused the inhibition of tertiary blood vessel formation in CAM assay.	[244]
Green synthesis using *Ficus religiosa*	5–35 nm	n. s.	spherical	Dalton’s ascites lymphoma mice model	Antiangiogenesis occurs afer AgNPs treatment of DAL tumor cells. Anti-angiogenic activity was confirmed by observing vessel development.	[245]
Biosynthesis using *Aspergillus niger SAP2211*	SEM 9.2–50 nm	SEM 13.53 ± 4.08 nm	SEM spherical, oval	*in ovo* CAM inoculated with HeLa cells	AgNPs reduced intercapillary network and inhibited the angiogenesis. No significant changes were observed in AgNPs-treated normal CAM, but the synthesized biogenic AgNPs showed a dose-dependent significant decrease in vascularization of CAM inoculated by HeLa cells.	[246]
	TEM 8–55 nm	TEM 30.31 ± 3.36 nm	TEM different morphologies			
Biosynthesis using *Bacillus licheniformis*	40–50 nm	50 nm	spherical	porcine retinal endothelial cells	AgNPs show inhibitory effect on the vascular permeability induced by VEGF and IL-1β through inactivation of Src kinase pathway. AgNPs blocked the Src-phosphorylation at Y419.	[228]
Biosynthesis using *Bacillus licheniformis*	40–50 nm	50 nm	spherical	bovine retinal endothelial cells	500 nM of AgNPs inhibit VEGF and IL-1β-induced cell proliferation and migration.	[230]
					500 nM silver nanoparticles decreased viability to 50% of initial via PI3K/Akt-dependent pathway. Inactivation of Akt by silver nanoparticles is associated with caspase-3 activation and DNA fragmentation. It means that Ag NPs activated the process of apoptosis.	
Biosynthesis using *Bacillus licheniformis*	40–50 nm	50 nm	spherical	bovine retinal endothelial cells such as PEDF	AgNPs could inhibit VEGF, cell proliferation, migration, and capillary-like tube formation.	[229]
				C57BL/6 mouse Matrigel plug assay	AgNPs strongly inhibited the vessel number and the formation of new blood microvessels induced by VEGF. AgNPs could inhibit the activation of PI3K/Akt signaling pathways.	
Plumbagin caged silver nanoparticles	n. s.	150 nm	face-centered cubic crystalline nature	endothelial tube formation assay	AgNPs exhibited poor tube formation and absence of cell–cell contact under VEGF stimulated condition.	[238]
				CAM	AgNPs under VEGF-stimulated conditions reduced number of blood vessels and so inhibited angiogenesis.	
				chick aorting ring sprout formation assay	AgNPs reduced sprout formation.	
				endothelial cells	AgNPs modulated the VEGF activity by PAR modification which leads to inhibition of angiogenesis.	
				bovine retinal endothelial cells	AgNPs could inhibit the VEGF-induced migration, proliferation, and capillary-like tube formation	
				mouse Matrigel plug assay	AgNPs effectively inhibited the formation of new blood microvessels induced by VEGF	
Reduction in AgNO_3_ with diaminopyridinyl-derivatized heparin polysaccharides (Ag DAPHP)	10–30 nm	n. s.	polydisperse	CAM	AgNPs inhibited FGF-2-induced angiogenesis with an enhanced antiangiogenesis efficacy with the conjugation to Ag DAPHP as compared with glucose conjugation. AgNPs enhanced the antiangiogenic effects compared with DAPHP alone and glucose-reduced AgNPs. Free Ags are toxic, but glucose and HP impart biocompatibility to these particles. Conjugated Ag to HP or HA showed improved antiangiogenesis efficacy as compared with Ag–glucose.	[231]
				mouse matrigel model	AgNPs shown antiangiogenesis efficacy in the FGF-2 mouse Matrigel and CAM model.	
Obtained from Nanjing XFNANO Materials Tech Co. Ltd.	10 nm	n. s.	spherical	MCF7, HeLa, human primary chondrocytes, and the human lymphoma cell lines Raji and Daudi	AgNPs attenuate HIF-1α and HIF-2α accumulation and suppress the transcriptional activity of HIF, resulting in inhibition of the expression of downstream target genes VEGF-A and GLUT1. The consequence is also suppression of tube formation during angiogenesis. AgNPs also interfered with the accumulation of HIF-1α protein and the induction of the endogenous HIF target genes, VEGF-A and GLUT1. AgNPs disrupt HIF signaling pathway and downstream VEGF-A function.	[227]
				Matrigel	400 µg/mL AgNP completely inhibited tube formation of HUVEC on Matrigel. AgNPs block in vitro angiogenesis by inhibiting tube formation.	
**Pro-angiogenic effect**						
Polyvinylpyrrolidone (PVP)-coated AgNPs	n. s.	2.3 nm	spherical	SVEC4-10 cells	AgNPs induced production of ROS, production of angiogenic factors (VEGF and nitric oxide, support the activation of Akt, FAK, p38, ERK1/2, and p38, connected with VEGF-receptor-mediated signaling pathway.	[242]
				B16F10 melanomas injected into C57BL/6 mice	AgNP caused, in a concentration-dependent manner, increasing angiogenesis around tumors and the content of hemoglobin.	
				C57BL/6 mouse Matrigel plug assay	AgNP-containing Matrigel stimulated angiogenesis and increased the infiltration of endothelial cells and hemoglobin content.	

**Table 4 life-13-00466-t004:** Some recent studies evaluating the potential of AgNPs as in vivo anticancer agents.

AgNPs Features		Exposure Duration	Tumor Model	Antitumor Effects	Comments and References
*Size* *(nm)*	*Surface Coating*				
n.r.	Biogenic *(F. oxysporum)*	3 weeks of I.Ve doses	Bladder cancer-induced mice	0.05 mg/mL led to 57% tumor regression.	[252]
74	Citrate	I.P injection twice a week	CT26 tumor-induced in mouse models	2 mg/kg twice a week (bw) led to a significant decrease in tumor growth kinetics compared with nontreated control animals.	Treatment with AgNPs resulted in temporary (>60 days) remission of CT26 tumors [253]
9–25	Biogenic *(N. linckia* pigment extract)	10 days daily I.P injections	EAC tumor-induced mouse model	5 mg/kg (bw) inhibited tumor development (volume, number of tumor cells, and weight).	No histopathological alterations in major organs (liver, spleen, and kidney) after treatment [254]
50	Biogenic (*B. licheniformis*)	15 days I.P injection	DLA tumor–induced mouse model	500 nM led to a reduction in DLA cell count.	Increase in survival time by 50% [247]
25	PVP	10 weeks I.V injection 3 times/week	TNBC tumor-induced mice model	6 mg/kg (bw) caused significant tumor development.	100% survival rate in AgNP-treated mice and only 30% in control groups [110]

*n.r*: Data not reported by the authors.

## Data Availability

Not applicable.

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
