# Peer review of "The Role of Silver Nanoparticles in the Diagnosis and Treatment of Cancer: Are There Any Perspectives for the Future?"

_life, 2023, doi:10.3390/life13020466_

Round 1

Reviewer 1 Report

I have read the manuscript entitled "The role of silver nanoparticles in the diagnosis and treatment of cancer: are there any perspectives for the future?" with great interest and I think it is in principle suited for a publication in the Life, Special Issue “Silver Nanoparticles Synthesis, Characterization, Biological Activities”. The manuscript brings some new information to the scientific community. I am happy that tables and figures are well presented in general. However, I also have concerns and comments.

Comments:

Page 2. I think that AgNPs are not discussed in the cited article #16 (maybe, https://doi.org/10.1016/j.actbio.2010.08.003 is appreciated).

Table 1. In the table 1 methods of AgNPs synthesis by algae, spray pyrolysis nanolithography are just mentioned but not described and discussed in the manuscript. Please check and describe.

Page 4. “The sonochemical approach is safe, and fast with good control of reaction conditions [35–39].” I think that sonochemical approach is not used in the cited work #35. Please check.

Page 4. “Useful biological sources are mainly plant extracts, but also microorganisms and biomolecules, such as saccharides, polysaccharides, proteins, and amino acids [43–48].” I think that nanoparticles are not discussed in the cited work #47. Please check.

Page 4. “As we mentioned in Table 1, biological methods can be divided into in vivo and in vitro, hence intracellular and extracellular.” I think that "in vitro" methods are not equal to “extracellular” (e.g. https://doi.org/10.1186/s12989-018-0278-9). I think also in the Table 1 biological methods can be divided into intracellular and extracellular (e.g. Figure2 in https://doi.org/10.3390/jof7040291).

Page 4. “Other authors also successfully synthesised AgNPs by living plants Festuca rubra, Medicago sativa, and Brassica juncea [52]”. I think that the plants are not discussed in the cited work #52. Please check.

Page 4. “The substrate participation from the initialization of the growth increases the activity of enzymes [43, 53–56]” I think that the statement is not discussed in the cited papers ##53-56. Please check.

Pages 4-5. “Since that time some more reports have focused on the biosynthesis of AgNPs by microorganisms, namely Bacillus methylotrophicus [58], Lactobacillus strain [59,60], or E. coli [61]” I think that Lactobacillus strain is not discussed in the cited work #60. 

Page 8. “The dynamic light scattering (DLS) method plays its role in size distribution measurements of small particles in solution or suspension [100,102].” I think DLS is not discussed in the cited article #100. Please check.

Page 10. “Blood-dispersed silver nanoparticles are encased in a unique protein covering, the so-called "Protein crown" (PC). Changes in protein patterns are difficult to detect with traditional blood tests; nevertheless, PC functions as a "nanoconcentrator" of serum proteins with affinity for the surface of AgNP.” Please add relevant reference.

Page 10. “Oraevsky was the first to propose photoacoustic imaging for biological purposes [117]”. Please cite original work.

Page 10. The subhead “AgNPs as cell cycle modulators and inhibitors of p53 regulation”. I think that p53 is just mentioned but not described and discussed in this subhead (Pages 10-11). The next subhead is "Effect of AgNPs on cell cycle regulation".

Page 15. “The result of such mitochondrial damage by AgNPs is an increase in the number of cells with reduced MMP [135, 153]”. What is  the "MMP" (mitochondrial membrane potential, mitochondrial outer membrane permeabilization or matrix metalloproteinase)? Please describe.

Pages 15-16. “It was found that these NPs were able to activate the PERK-mediated signaling pathway”. Please add relevant reference.

Page 17. The section "Non-apoptotic cell death". I think that non-apoptotic cell death not described and discussed in the cited work #192. Please check.

Page 18. “In contrast, AgNPs biogenerated by Klebsiella oxytoca under anaerobic conditions induced autophagic cell death in SKBR-3 (human breast adenocarcinoma) cells”. Please add relevant reference (e.g. #129).

Table 2. Line "Juniperus…” Please check the reference #25 (#27 maybe is correct).

Table 2. Line “Zizyphus mauritiana … “ Please check the reference #151 (#149 maybe is correct).

Table 2. Line “AgNPs … [160]”. Maybe delete "AgNPs" in first column? 

Table 3. Line “Daphnia magna juveniles AgNPs (S, 23 nm and 27 nm) [249]” I think that human lung carcinoma cells (A549) and Ag–In–Zn–S quantum dots are studied in the cited article #249. Please check. 

Table 3. Line “Chemosenzitizing effect Colo 320 Citrate-coated AgNPs (S, 28 nm) [248]”. Please check the reference #248 (#245 maybe is correct).

Table 3. I think that HepG2 are not studied in the cited work #250.

Table 3. Reference #251. Also I think that HeLa cells are not studied in the cited article. Please check it.

Page 34. “Although it has been discovered that AgNPs suppress tumour invasion [270]”. Please check relevance of the cited article.

Page 37. “In addition, it has been shown that AgNPs and the topoisomerase I inhibitor camptothecin promote cervical cancer cell death in a synergistic manner”. Please add relevant reference.

Page 38. “AgNPs inhibited the production of cytokines, including IFN-γ, IL-6, IL-8, IL-11, TNF-α, and, to a lesser extent, IL-5, in human peripheral blood mononuclear cells and human mesenchymal stem cells”. Please add relevant reference.

Figure 10. “Stress fibres” . Please check (maybe “stress fibers”).

Page 40. “and was first hypothesised by Folkman (1971) more than 50 years ago [324]”. Please add original reference.

Page 41. “such as human um-bilical vein endothelial cells (HUVEC) [330]”. Please check the reference #330 (#329 maybe is correct).

Page 41. “bovine retinal endothelial cells [339]”. Please check the reference #339. (#331 maybe is correct).

Page 41. “bioartificial renal epithelial cells (BREC) [331]” I think that bovine retinal endothelial cells are not tested in the cited article #331.

Page 41. “Various animal models can be used, for example the in ovo or ex ovo chick embryo chorioallantic membrane (CAM) assay [331, 332], rat aortic ring model [334], rabbit cornea, mouse matrigel implant assay [333] or C57BL/6 mice [331].” Please check references.

Page 45. “In a research study by Sriram et al. [347]”. Please check the authors “Sriram et al.” in References.

Page 46. “treatment with the anti-cancer medicine. 5-Fluorouracil (20 micrograms per kilogramme).” Please check.

Page 46. “The levels of p53, p21, and cleaved caspase-3 rose in the liver tissue of male Sprague Dawley rats given orally with up to 100 mg/kg/day PVP-coated nanosilver (20–30 nm) for 90 days, then declined at higher doses where autophagic cell death was believed to occur [353].” I think that the liver tissue of male Sprague Dawley rats is not studied in the cited work #353.

Page 46. “and metastatic dissemination of 4T1 tumours in mice [359]”. I think that "4T1 tumours" are not studied in the cited work #359.

Page 49. “and impairment of platelet function [91,3,301]”. Please  check relevance of references.

Page 51. "using the comet assay when treated with non-toxic levels of nanosilver (10 and 40 nm; citrate-coated) [297]”. Please check relevance of the reference #297.

Best regards.

Reviewer 2 Report

Although the review are well organized but it its length may be in book chapter not the review article

Reviewer 3 Report

Authors reviewed the role of silver NPs (AgNPs) in cancer nanomedicine. They discussed numerous mechanisms which render anticancer properties under both in vitro and in vivo conditions, as well as their potential in the diagnosis of cancer. n the diagnosis of cancer. 

The manuscript is well written and suitable for publication in life. 

Reviewer 4 Report

General comments

The present manuscript focuses on reviewed the role of silver NPs (AgNPs) in cancer nanomedicine, discussing numerous mechanisms by which they render anticancer properties under both in vitro and in vivo conditions, as well as their potential in the diagnosis of cancer. The manuscript sounds scientific and a comprehensive knowledge of the AgNP-induced effects on single cells, cancer tissues, and organs as well as their potential in cancer diagnosis was discussed. The beginning of the article is devoted to the synthesis and characterization of silver nanoparticles as well. However some points are suggested to improve the overall quality of the manuscript before final publication.

Moderate English editing is required.

General comments and suggestions for authors:

Un-bold the sentence: “Silver nanoparticles synthesis is usually divided into two main groups of approaches – top-down (including physical synthesis) and bottom-up methods (chemical and biological synthesis) (Table 1)”.

Un-bold Table 1 contents.

Strictly follow the journals pattern for references and layout.

Some sections and sub-sections are without numerical order. Add them chronological order.

Figures: the figures must have high resolution. These should be modified.

Abbreviations should not be added in the middle of the text. They should either be in the first section or should be in the end of the manuscript as per the journal’s format.

The review is very length. Some irrelevant information is added which is quite disoriented.

Conclusion: It should be precise and summarize the overall progress with some future perspectives.

Reviewer 5 Report

Quick comments for the Authors:

Given the large number of experimental techniques, and interaction methods of AgNPs described in the present review, I would suggest the Authors to add a sort of table of contents between the abstract and the introduction paragraphs. Indeed, paragraph numbering stops right after section "Characterization techniques". Moreover, subparagraph numbering may be considered as well. This helps the reader a lot when in search of a specific topic within the review work.

Reviewer 6 Report

The authors drafted a comprehensive review about preparation, characterization, and bio-related application of silver nanoparticles. The referee believes the manuscript is suitable for publication, and a minor issue can be addressed before publication, to improve the quality of the review.

There has been a lot of reports of ligand-capped silver nanoparticles. As the ligands can provide improved stability and functionality, the referee believe it would be better if the authors could include a few references to reflect the current progress in this direction.

Round 2

Reviewer 1 Report

The manuscript entitled "The role of silver nanoparticles in the diagnosis and treatment of cancer: are there any perspectives for the future?” has been significantly improved and I can suggest that manuscript can be published after minor text editing.

Minor comments:
Table of contents. Please check the heading numbers.

Page 22. "Tu dať pár viet z článku autorov Fageria et al. 2017."  Please delete.

Best regards.

Reviewer 2 Report

The manuscript needs a major revision of English errors and add more recent topics to update the main goal of the review article.
